# Probing the modulation of enzyme kinetics by multi-temperature, time-resolved serial crystallography

Eike C. Schulz [1,2,3,6] ✉, Andreas Prester [1], David von Stetten [4], Gargi Gore [1], Caitlin E. Hatton [3], Kim Bartels[1], Jan-Philipp Leimkohl [2], Hendrik Schikora[2], Helen M. Ginn[5], Friedjof Tellkamp [2,6] ✉ & Pedram Mehrabi [2,3,6] ✉

The vast majority of protein structures are determined at cryogenic temperatures, which are far from physiological conditions. Nevertheless, it is well established that temperature is an essential thermodynamic parameter for understanding the conformational dynamics and functionality of proteins in their native environments. Time-resolved crystallography is a technique that aims to elucidate protein function by examining structural alterations during processes such as ligand binding, catalysis, or allostery. However, this approach is typically conducted under ambient conditions, which may obscure crucial conformational states, that are only visible at physiological temperatures. In this study, we directly address the interplay between protein structure and activity via a method that enables multi-temperature, time-resolved serial crystallography experiments in a temperature window from below 10 °C to above 70 °C. Via this 5D-SSX, time-resolved experiments can now be carried out at physiological temperatures and with long time delays, providing insights into protein function and enzyme catalysis. Our findings demonstrate the temperature-dependent modulation of turnover kinetics for the mesophilic β-lactamase CTX-M-14 and the thermophilic enzyme xylose isomerase, within the full protein structure

It is well established that temperature plays a critical role in protein function, significantly influencing structural dynamics, ligand binding, catalytic rates of enzymes, and allosteric interactions[1–3]. This interplay can be probed by several methods, such as nuclear magentic resonance (NMR) spectroscopy, isothermal titration and differential scanning calorimety (ITC / DSC) as well as molecular dynamics simulations[1]. However, to reduce the rate of radiation damage, the vast majority of protein structures has been acquired at cryogenic temperatures[4]. In this context, it is important to note that flash-cooling may introduce structural artefacts, which are absent at ambient or physiological temperatures, that may affect biological interpretation[5,6]. It has been shown that cryo-cooling may remodel a large proportion of side-chains and thereby provide a picture of unrealistic protein structures that hide functionally important conformations present at physiological temperatures[7–9]. In light of structure-based drug design campaigns it is of great concern that temperature can affect the structure of protein ligand complexes[10–13]. These effects are not limited to extreme temperature differences but can be a critical determinant in ligand binding when comparing room-temperature to physiological temperatures[14]. However, temperature dependent structural differences also include

[1]University Medical Center Hamburg-Eppendorf (UKE), Hamburg, Germany. [2]Max-Planck-Institute for the Structure and Dynamics of Matter, Hamburg, Germany. [3]Institute for Nanostructure and Solid State Physics, University of Hamburg, Hamburg, Germany. [4]European Molecular Biology Laboratory (EMBL), Hamburg, Germany. [5]Deutsches Elektronen Synchrotron (DESY), Hamburg, Germany. [6]These authors contributed equally: Eike C. Schulz, Friedjof Tellkamp, Pedram Mehrabi. ✉e-mail: ec.schulz@uke.de; friedjof.tellkamp@mpsd.mpg.de; pedram.mehrabi@uni-hamburg.de

the overall hydrogen bond network, which includes the hydration shell of the protein. Recent reports highlight the influence of temperature on the structure of the water network and its ability to alter protein conformation and ligand interactions[15,16]. This is of importance as proteins do not obtain a single unique conformation, but rather fluctuate around an energetic ground-state. Consequently, these conformational dynamics are again intimately linked to temperature, the protein hydration shell, and ligand binding events with direct implications for cooperativity, allostery and catalytic efficiency of enzymes[2,17–21]. Fraser and colleagues have shown that the conformational ensemble includes functionally relevant, temperature-dependent conformations that may not be observable at lower temperatures[22]. Therefore, experimental structures that closely resemble physiological conditions are becoming increasingly significant, especially in light of the recent feasibility to computationally predict protein structures[23–27]. Thus, the mismatch between data collection temperature and the protein's physiological activity temperature, displays a major gap in our understanding of protein function.

This dilemma is particularly important in the context of time-resolved structural studies that seek to provide detailed insight into catalytic mechanisms and allosteric regulation. In spite of this challenge, the majority of time-resolved structural studies are carried out at ambient temperatures, regardless of the actual physiological or optimal temperature of the protein under study[28,29]. While the equilibrium structure of the conformational ensemble can be addressed at different temperatures, via NMR spectroscopy or multi-temperature X-ray crystallography, these have remained niche applications in comparison to the majority of structure determinations. Although a key environmental variable for mechanistic time-resolved analyses, considerable technological difficulties have so far prevented routine use of physiologically relevant temperatures during time-resolved structural studies. To address temperature-dependent, out-of-equilibrium conformations, pioneering single-crystal work had been carried out on the reversible system photoactive yellow protein (PYP)[30,31]. More recently, temperature-jump serial crystallography addressed the evolution of the conformational ensemble in response to an infra-red laser pulse, which is used to externally trigger molecular motions, to map protein dynamics as the system relaxes to a new equilibrium[32].

Here we present a novel method that enables multi-temperature time-resolved serial crystallography (5D-SSX) experiments. Our primary objective is to specifically address the interplay between protein structure, dynamics, and activity as a function of temperature. By maintaining the protein crystals at a defined temperature and humidity, our method enables studying molecular motions in response to physiological events, especially at comparably long time-delays, which are often encountered during enzyme catalysis[33]. Especially the effects of relative humidity on the crystals' unit cell have been noted early on by Perutz and Kendrew, who noted swelling or shrinkage of hemoglobin crystals[34,35]. This was later used to tailor the diffraction properties of single protein crystals[36–39]. Via our environmental control box this is now accessible to serial crystallography, too. The implementation of serial crystallography permits to conveniently address irreversible enzymatic reactions that escape time-resolved analyses within single macroscopic crystals, and our *hit-and-return* (HARE) approach is especially advantageous to address the long biologically relevant delays times[40]. Moreover, employing a serial crystallographic approach also has clear advantages with respect to radiation damage, as the total dose is distributed over several thousand individual crystals[41,42]. Additionally, we utilise our *liquid-application method for time-resolved analyses* (LAMA)[43]. This enables to also address the predominant fraction of enzymes that are not natively photoactivce[44], but rather using the most simple way of reaction initiation by directly mixing soluble ligands with the protein crystals[45]. These features render our method particularly versatile and enable addressing a wide spectrum of different systems.

To demonstrate the versatility of the method, we have employed two different model systems: a mesophilic enzyme with direct biomedical relevance[46], and a thermophilic enzyme with industrial implications[47,48].

Extended spectrum class-A serine $\beta$-lactamases (ESBL, CTX-M, EC:3.5.2.6) have been identified in a growing number of clinical isolates of Gram-negative bacteria, with worldwide distribution[46,49]. *Klebsiella pneumoniae* CTX-M-14 is a monomeric enzyme that catalyzes the hydrolysis of penicillin-, cephem- and carbapenem-family antibiotics[46,50]. Canonically, a carboxylate or negatively charged group on the $\beta$-lactam antibiotic binds to positively charged residues in the active site. This is followed by acylation of the $\beta$-lactam ring, which subsequently forms a covalent bond with the catalytic serine residue. Subsequent to this, an activated water molecule hydrolyses the acyl-enzyme intermediate, leading to the release of the ring-opened, inactivated antibiotic[49,51]. Previous reports on in-solution kinetics vary considerably depending on substrate and reaction conditions[52]. For instance, the turnover constants ($k_{cat}$) for CTX-M-14 with cefotaxime demonstrated $k_{cat}$ values between 37 s$^{-1}$ and 1400 s$^{-1}$ [46,53–55]. This prompted us to test a variety of alternative substrates, from which we selected piperacillin as the model compound for this study.

*Streptomyces rubiginosus* xylose isomerase (XI, EC:5.3.1.5), is a tetrameric enzyme that facilitates the interconversion of aldoses to ketoses, for example D-xylose into D-xylulose or D-glucose into D-fructose. Its catalytic reaction depends on two divalent metal ions and a histidine residue in the active site. The initial formation of an open-ring intermediate is followed by its conversion into a 5-membered ketose ring via a hydride shift mechanism[56,57]. The in-solution $k_{cat}$ of XI has previously been determined to be 53 s$^{-1}$ at an optimal reaction temperature of approximately 80 °C (pH 7.5). In contrast, at 60 °C (pH 7.3) the $k_{cat}$ for glucose has been reported to be approximately 5.3 s$^{-1}$, while at 35 °C, XI has been reported to be nearly inactive[58–62].

Here, we show for both systems that a modification of the environmental temperature modulates the underlying structural dynamics as well as the enzyme kinetics. This enables us to resolve alternate structural intermediates that would either be convoluted with other conformational states or could not be observed at high occupancy at ambient temperatures.

## Results and Discussion
### Environmental control enables multi-dimensional SSX
In order to facilitate 5D-crystallography[30] experiments, we have developed a modular environmental control system. For a detailed description of the environmental control box design and characterisation of the temperature and humidity controls, please refer to the supplementary material. Briefly, the environmental control box consists of a rail-mounted, retractable, perspex housing with cyclo-olefin copolymer (COC) and mylar X-ray inlet and outlet windows, respectively (Fig. 1a). Constant relative humidity and temperature can be achieved by two independent closed loop control circuits, whereby the former is controlled by a flow of humidified air from a hot water bath, that enables precise control of the relative humidity to within a percent. Two interchangeable modules are used to control the temperature. Module-1 contains water-cooled Peltier elements, while module-2 consists of simple heating resistors. The modules cover the ranges from 7 °C to 55 °C and 50 °C to above 70 °C, respectively. This temperature window is large enough to encompass physiologically relevant temperatures of both mesophile and hyperthermophile enzymes. Protein micro-crystals are loaded onto our fixed-target 'HARE' chips, and raster-scanned through the X-ray beam[63]. Via full compatibility with our previously described HARE, and LAMA methods, this system permits multi-temperature, time-resolved serial crystallography experiments for versatile delay times[40,43].

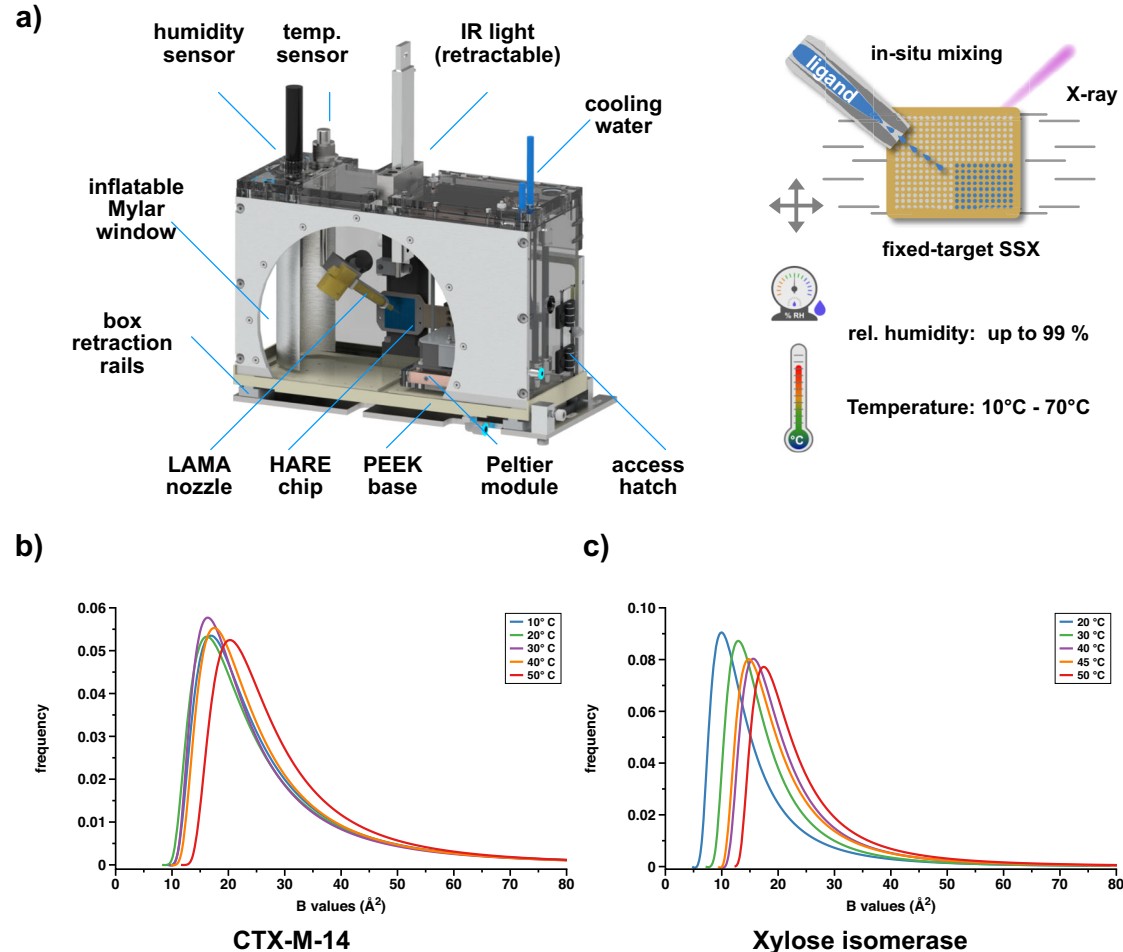

**Fig. 1 | The environmental control box enables recording multi-temperature serial crystallography data. a** The environmental control box, the portal translation for the LAMA nozzle is hidden for clarity (further details in the supplementary notes). **b, c** ADPs of CTX-M-14 and XI for models derived from data recorded at different temperatures, fitted to a shifted inverse gamma distribution. Source data are provided as a Source Data file.

**Temperature modulates structural dynamics in protein crystals**

In order to establish a reliable and accurate method for determining the influence of temperature on the resting state of the CTX-M-14, and XI structures, initial data were collected without triggering a reaction. After equilibration of the respective temperatures, all parameters were kept equivalent between chips (humidity, beamline, data collection), and structures were determined using crystals from the same crystallisation batch. Five structures of CTX-M-14 were determined at 10 °C, 20 °C, 30 °C, 40 °C, and 50 °C, as well as five structures of XI at 20 °C, 30 °C, 40 °C, 45 °C, and 50 °C (Supplementary Tables 1, 2). This systematic normalization of all experimental parameters permits a direct side-by-side comparison of the structures and consequently allows to directly assign temperature-induced structural differences. To assess the global variability of atomic positions in a comparable way, we determined the atomic displacement parameters (ADP), and fitted these to a shifted inverse gamma distribution (SIGD) as a function of temperature[64] (Fig. 1b, c). To correlate the global structural differences seen in the SIGD with an in-solution behaviour of our model systems, we determined their melting points *via* differential scanning fluorometry (Supplementary Fig. 1a,b) in four different buffer systems, also taking the crystallization condition into account (Table 1). The melting point of CTX-M-14 (ca. 53 °C) is in good agreement with previous reports[51] and matches the displacement of the SIGD for the 50 °C SSX structure. By contrast, the multi-temperature SSX experiments for XI do not reach the melting point of XI (ca. 83 °C), but continuously increasing ADPs of XI can be observed in the SIGD. In order to quantify

and visualise the changing global structural differences, a pairwise backbone Root Mean Square Deviation (C$\alpha$-RMSD) was calculated for both model systems and represented as a categorical heatmap (Supplementary Fig. 1c, d), showing temperature-dependent clustering for both model systems. For CTX-M-14 a shift in the ADP distribution and pairwise C$_\alpha$-RMSD can only be determined above a threshold temperature (ca. 50 °C), corresponding to its in-solution melting temperature (Fig. 1b, Supplementary Fig. 1). In contrast, XI exhibits a substantially elevated in-solution melting temperature and shows a monotonous ADP increase with higher temperatures (Fig. 1c, Supplementary Fig. 1). However, ADPs do not only display local structural dynamics but also contain other sources of hierarchical disorder, such as lattice disorder and molecular displacements[65], and may thus disguise subtle conformational changes. As changes in torsion angles are separated by lower energy barriers than changes in coordinates, they are more sensitive to conformational dynamics. Previously, it was established that torsion angle deviations preserve temperature induced dynamics[66]. Therefore, we analysed the torsion angle distribution in both enzymes, to further investigate the response to altered environmental temperatures and delineate local structural changes. Briefly, the software RoPE begins by enforcing canonical bond length and angle geometry on every structure, and optimising torsion angles to account for this[67]. For each bond, the mean torsion angle is calculated from all structures. For each structure, a list of torsion angle deviations from the mean is compiled. For every pair of structures, a correlation coefficient between these torsion angle

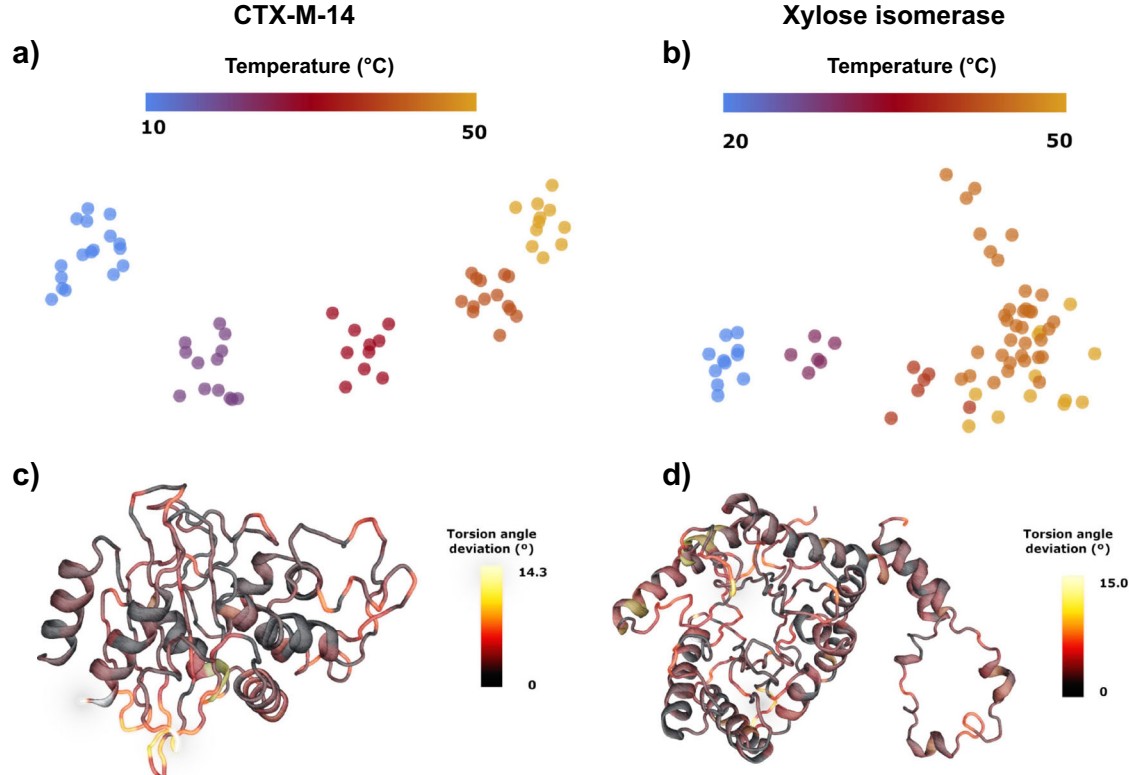

**Fig. 2 | Torsion angle dynamics analysis. a, b** Atomic coordinate-based RoPE space for both proteins at each temperature[66]. Each point corresponds to an experimentally determined structure based on a subset of diffraction images, showing the first two principal components. **c, d** Magnitude in torsion angle deviation corresponding to the first principal component of (**a**) and (**b**), respectively, plotted as a heatmap onto the backbone of the structures, illustrating the local structural response to temperature changes.

deviations is generated. Singular value decomposition (SVD) is carried out on the full correlation matrix representing all structure pairs. RoPE then plots a RoPE space, transforming each structure into the coordinates represented by the **U** matrix. Three-dimensional plots represent the three highest ranked vectors of the subspace, rotated to maximise clarity, and projected onto 2D for the figure display. Therefore, these plots show the inherent variation between structural conformations as determined from their torsion angles. For the RoPE analysis, the datasets were split into sub-datasets based on at least 2000 diffraction patterns. Clearly, both the CTX-M-14 as well as the XI sub-datasets assemble into temperature-specific clusters, emphasising a coherent conformational space that corresponds to environmental temperature (Fig. 2). An interesting deviation from this trend can be seen in the XI 45 °C structure, in RoPE-space, SIGD, as well as in the pairwise $C_\alpha$-RMSD, indicating a general structural response in this structure. Taking the general trend into account, the combination of the much higher unfolding temperature of XI in solution, and the gradual increase in ADPs with increasing temperature suggests a different temperature response mechanism of the hyperthermophile XI compared to the mesophilic CTX-M-14. This indicates that the respective response in conformational variability reflects an adaptation to either a mesophilic or a thermophilic temperature optimum, respectively. In other words, the higher flexibility observed for XI serves to accommodate the higher degree of molecular motions concomitant with higher temperatures, without compromising its functionality[48,58]. However, analysis of the torsion angle dynamics also permits pinpointing those structural elements that directly respond to temperature changes. Thus, mapping the torsion angle deviation onto the backbone of the protein structures highlights hinge regions with particular flexibility (Fig. 2). In CTX-M-14 these are residues 52-54, 193-196, 226-231 and 251-255, while in XI the region around 25-26, 126-129

and 206-210 shows the largest torsion angle deviation. Collectively, these data unambiguously show a direct response of the conformational dynamics to environmental temperature, and that these temperature changes can be effectively projected to structural changes.

## Multi-temperature time-resolved crystallography (5D-SSX)
Next, we analysed the effect of temperature on the catalytic activity of our two model enzymes. To assess thermal effects on catalysis, we equilibrated the crystals at a given temperature and triggered their reaction by adding the substrate solution *via* the LAMA method[43] and monitored turnover at a constant time-delay of 3 s (CTX-M-14) and 60 s (XI) after reaction initiation, respectively (Supplementary Tables 3 and 4). These delay times were selected from temperature-resolved time-series of either enzyme, to enable a cross-temperature comparison. For an unambiguous assessment of the changes, we utilised $F_{obs}$-$F_{obs}$ Difference Electron Density (DED) maps, as well as absolute value Electron Number Density (END) maps, which are quantitatively comparable and therefore permit a side-by-side comparison of the different structures. However, it should be noted that any time-resolved crystallographic analysis will result in a superimposed mixture of sub-states, generating an ensemble of structures with respective fractional occupancies that contribute to the same electron density[68]. In order to address this situation and minimize interpretation bias, we initially assembled all previously known stable intermediates into one structure and relied on constrained group refinement to determine the occupancy of the different overlapping states at the respective datasets[69].

**Increasing temperature modulates substrate diffusion.** For CTX-M-14 the 20 °C structure corresponds to the unbound state, exhibited by rather discontinuous difference electron density and correspondingly

low occupancy for any of the piperacillin intermediates. The 30 °C structure on the other hand is clearly populated by difference density in the active site, corresponding to ligand molecules. Occupancy refinement indicated a mixture corresponding predominantly to the acyl-enzyme-intermediate and a complex with the hydrolysed piperacillin product. Finally, the 37 °C structure corresponds predominantly to CTX-M-14 in complex with the hydrolysed piperacillin product (Fig. 3a–c). These different catalytic states are further supported by alternate conformations of the catalytic serine (S70) that shifts its position during catalysis, as indicated by an altered population of the difference electron density (Fig. 3d). Interestingly, the conformation of the piperacillin hydrolysis product is clearly distinct from a previously reported complex in context of a mutant enzyme (S70G) (Supplementary Fig. 3). While the mutant structure (PDB-ID: 3Q1F) adopts a conformation akin to the acyl-enzyme intermediate, our product complex obtained in the wild-type protein and at physiological temperature is shifted between 1.8 Å and 3 Å from that conformation. Considering the overarching clinical relevance of ESBLs for bacterial infections, this previously unobserved product complex conformation emphasises the importance of detailed structural information under physiological conditions. The increase of electron density in the active site proportionally to temperature indicates and increase of substrate-diffusion with increasing temperature. Since different catalytic sub-states of CTX-M-14 can clearly be distinguished at different temperatures (unbound state, acyl-enzyme intermediate, hydrolysed piperacillin product Fig. 3a–c), these data also show that turnover kinetics of mesophilic enzymes can be modulated by temperature variation in our control box. Temperature modification can therefore become an important tool for the selective enrichment and differentiation of specific metastable or stable reaction intermediates, in the context of time-resolved serial crystallography.

**Increasing temperature modulates substrate turnover.** While the piperacillin hydrolysis by CTX-M-14 is irreversible and progressively proceeds towards a product-bound state, glucose to fructose conversion by XI can also proceed in the backward direction[46,48]. Accordingly, the system obtains an equilibrium between glucose and fructose over time. Snapshots along the reaction coordinate pathway would therefore reflect fractional occupancies of both species, mixed with open-ring reaction intermediates. We therefore modelled the unbound state, the Michaelis-Menten complex with closed-ring glucose as well as an open-ring glucose intermediate consistent with previously determined structures. The respective substrate complex structure at 20 °C can be superimposed with an RMSD of 0.1 Å to previously reported substrate complexes at room temperature (PDB-ID: 3KCL)[57] (Fig. 4). As XI has an activity optimum at ca. 80 °C[48,58], an increase of the temperature for a given delay time should shift the population towards the product side. In line with this hypothesis, the XI snapshots 60 s after reaction initiation show progressively decreasing occupancy for the glucose substrate and increasing occupancy for the open-ring intermediate with increasing temperature (Supplementary Fig. 2c). An unambiguous interpretation based on the electron density alone is difficult due to the multiple overlapping states, which mandates the use of more complex modelling and analysis protocols in future experiments[69–71]. However, as can be seen from the END and DED map analysis, different catalytic states (i.e., closed- and open-ring intermediates) can be detected at the same delay time as the temperature is increased. This shows that there is a direct correlation between the increase in the catalytic activity of XI and the rise in temperature. This observation suggests that thermal modulation can be used as a method to shift reaction equilibria, facilitating the identification of intricate mechanistic distinctions that depend on the energy of the system.

Next, we addressed the question of whether temperature is a critical parameter for the observation of certain reaction intermediates or whether this can also be compensated for by longer delay times at lower temperatures. To this end, we compared the XI active centre at a delay time of 180 s, at 20 °C and 50 °C, respectively (Fig. 5, Supplementary Table 5). In practice, this delay time should be long enough to accumulate reaction intermediates even for enzymes with low activity. However, in agreement with previous biochemical data showing that XI is almost inactive at room temperature, the density of the active centre at 20 °C corresponds predominantly to a closed-ring glucose molecule. Conversely, at 50 °C, the electron density in the active site predominantly corresponds to an open-ring intermediate, thereby validating our prior observation and underscoring the pivotal role of temperature in the observation of reaction intermediates that are not discernible within a reasonable time frame using conventional methods.

In summary, the results demonstrate the potential of temperature as a crucial factor in time-resolved crystallography, facilitating the selective enrichment and discrimination of reaction intermediates. Addressing enzyme catalysis at varying temperatures should therefore facilitate the simultaneous correlation of the mechanistic and thermodynamic aspects of protein function. We anticipate that conducting analogous experiments on other systems will enable a comprehensive understanding of how proteins exchange energy with their environment and how this is associated with conformational dynamics and turnover. Multi-dimensional analyses could play a role in experimentally determining free-energy landscapes of proteins in action and thus deriving their unique catalytic pathways[30]. Clearly, temperature variation enables the modulation of substrate diffusion and enzyme kinetics in protein crystals, thereby altering enzyme turnover and allowing for a more in-depth characterisation of enzymatic mechanisms. Multi-dimensional experiments can thereby contribute to the evolving role of structural biology in enabling a comprehensive understanding of conformational dynamics and its role in protein function in the future.

## Methods
### Environmental control system
All information that is required for reproduction of the hardware, including 3D-PDFs, CAD files, step-files, electronics, and a bill of materials can be obtained under the following https://doi.org/10.5281/zenodo.12758835.

### Protein purification and sample preparation
Xylose isomerase (Uniprot ID: P24300) was cloned into pET-24a(+). The construct was transformed into E. coli strain BL21 (DE3) Gold and grown in TB medium supplemented with 25 µg/mL kanamycin at 37 °C until an $OD_{600}$ of 1.0–1.2 was reached. Protein expression was induced by addition of 1 mM IPTG, and the cells were incubated further at 18 °C for 16 h. The cells were harvested by centrifugation (7000 × g, 15 min, 4 °C) and resuspended in lysis buffer (50 mM HEPES pH 7.5, 500 mM NaCl, 5% (v/v) glycerol, 5 mM imidazole, 5 units/mL DnaseI, and protease inhibitor). After the cells were lysed by sonication, undisrupted cells and debris was separated by centrifugation (30,000 × g, 1 h, 4 °C). The supernatant was applied to a 5 mL HisTrap FF (Cytiva), washed with wash buffer (50 mM HEPES pH 7.5, 500 mM NaCl, 5% (v/v) glycerol, 30 mM imidazole), and XI was eluted in elution buffer (50 mM HEPES pH 7.5, 500 mM NaCl, 5% (v/v) glycerol, 250 mM imidazole). Protein containing fractions were pooled and HRV-3C protease was added to the eluate (0.3 mg HRV-3C protease for material derived from a 1 L culture). The sample was dialysed overnight against SEC buffer (50 mM HEPES pH 7.5, 150 mM NaCl) at 4 °C. Negative IMAC was performed to recover the cleaved XI. The protein was concentrated to 3 mL using a 10,000 MWCO concentrator (Sartorius) and applied to a Superdex 200 HiLoad 16/60 column (Cytiva) equilibrated with SEC buffer. Fractions containing the protein were pooled and a buffer exchange into crystallization buffer (10 mM HEPES pH 7.5) was

**Δt = 3 s**

**20 °C**               **30 °C**               **37 °C**

a)

**2F_o - F_c σ level:**

| 1.0 | 1.5 | 2.0 | 2.5 | 3.0 | 3.5 |

b)

**2F_o - F_c absolute value electron number density (END) maps at 0.7 e⁻/Å³**

c)

**DED maps are shown at an r.m.s.d. of ± 2.7**

d)

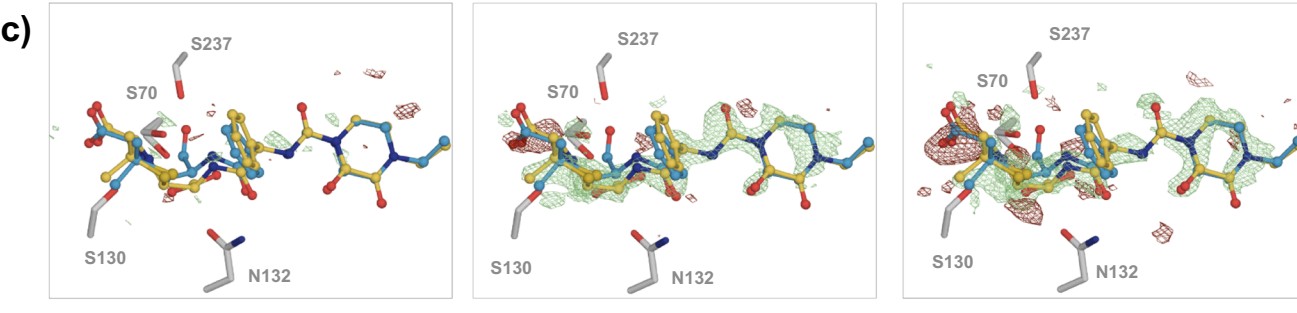

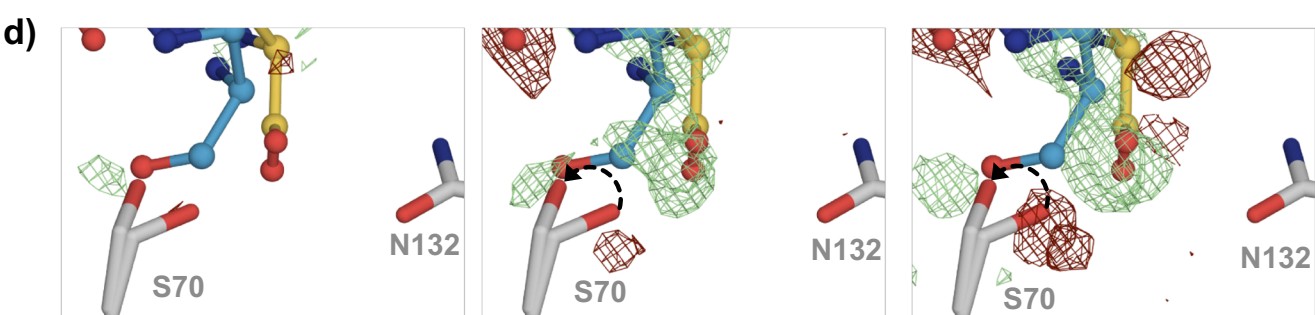

**Fig. 3 | Increasing temperature modulates substrate diffusion.** CTX-M-14 active site, 3 s after reaction initiation at 20 °C, 30 °C, and 37 °C; **a** $2F_o - F_c$ density shown at RMSD levels from 1.0–3.5, strongest density is shown in red; **b** $2F_o - F_c$ absolute value electron number (END) maps shown at 0.7 $e^-/\text{Å}^3$; c,d) $F_o - F_o$ difference electron density (DED) maps shown at 2.7 ± RMSD. **c** overview of the ligand density; **d** close-up of the catalytic Ser70. The piperacillin ligand is shown in the conformation of the covalent intermediate (blue) and the hydrolysis product (yellow), relevant CTX-M-14 active site residues are shown in grey, waters are omitted for clarity. All panels demonstrate the direct correlation between substrate diffusion and temperature as the electron density around the ligand increases relative to lower temperature. However, the DED maps also illustrate the change in relative occupancy, i.e., the concomitant conformational changes upon substrate hydrolysis.

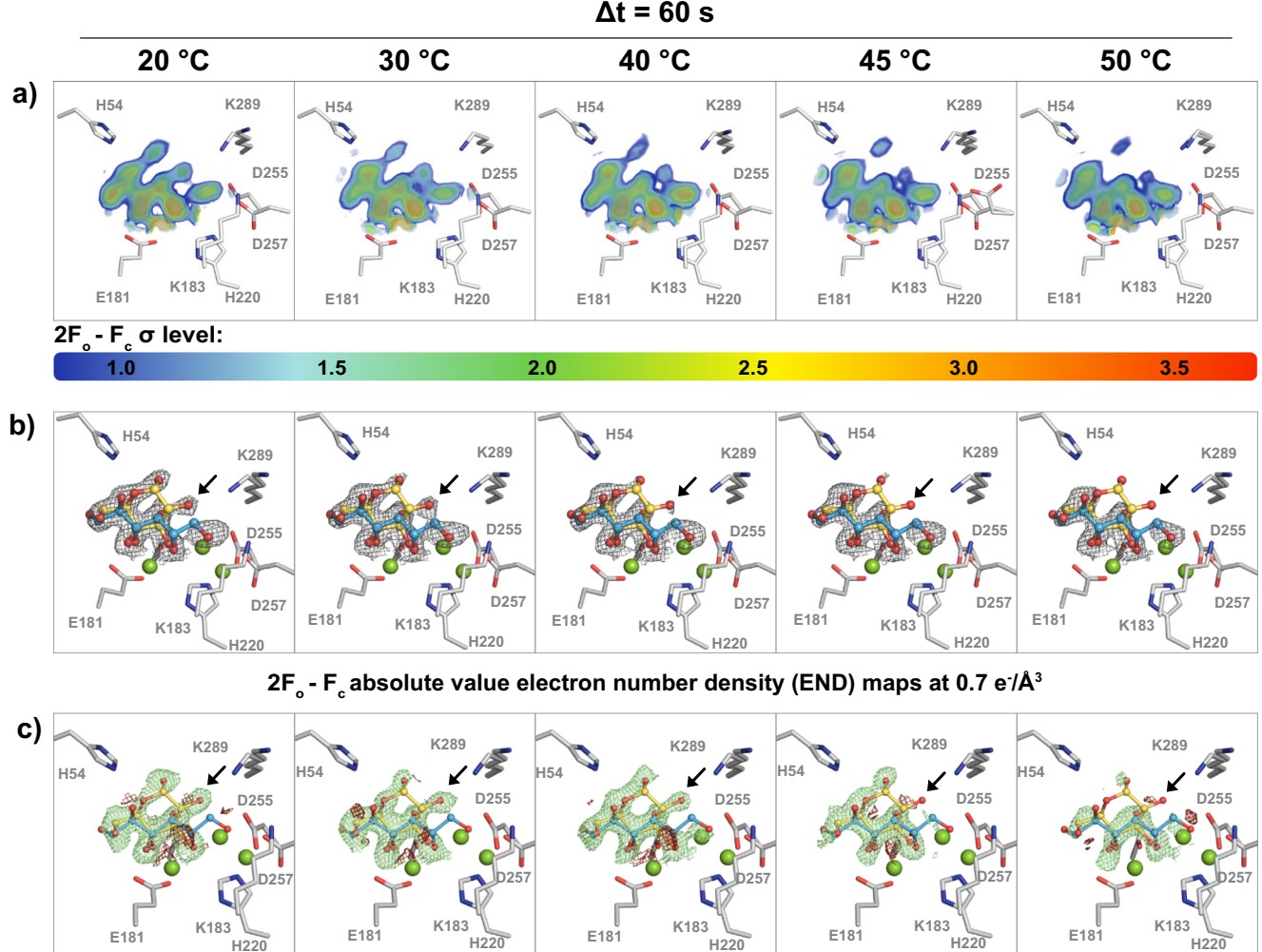

**Δt = 60 s**

**a)** 2F$_o$ - F$_c$ σ level shown for 20 °C, 30 °C, 40 °C, 45 °C, and 50 °C

2F$_o$ - F$_c$ σ level: 1.0 — 1.5 — 2.0 — 2.5 — 3.0 — 3.5

**b)** 2F$_o$ - F$_c$ absolute value electron number density (END) maps at 0.7 e$^-$/Å$^3$

**c)** DED maps are shown at an r.m.s.d. of ± 2.7

**Fig. 4 | Increasing temperature modulates substrate turnover.** XI active site, 60 s after reaction initiation at 20 °C, 30 °C, 40 °C, 45 °C, and 50 °C; **a** 2F$_o$ − F$_c$ density shown at RMSD levels from 1.0–3.5, strongest density is shown in red; **b** 2F$_o$ − F$_c$ absolute value electron number (END) maps shown at 0.7 e$^-$/Å$^3$, are quantitatively comparable and therefore enable a side-by-side comparison of the electron density, demonstrating its increase with raising temperature; **c** F$_o$ − F$_o$ difference electron density (DED) maps shown at 2.7 ± RMSD show the modulation of the ligand density. An arrow highlights the electron density change around the closed-ring glucose O2-oxygen. The glucose ligand is shown in closed ring conformation (yellow) and the open-ring intermediate (blue), relevant XI active site residues are shown in grey, waters are omitted for clarity.

performed using a PD10 column (Cytiva). For crystallization, the protein was then concentrated to 80 mg/ml using a 10,000 MWCO concentrator (Sartorius). Subsequently, microcrystals were obtained by vacuum induced crystallization, as originally described by Martin et al.[72], in XI crystallization buffer (35% (w/v) PEG 3350, 200 mM LiSO$_4$ and 10 mM Hepes/NaOH, pH 7.5). Sufficient microcrystal for a typical HARE chip were prepared from 25 µl protein solution (80 mg/ml) combined with 25 µl crystallization buffer. For droplet injection 1 M glucose solution was prepared in ddH$_2$O prior to data collection and stored at room temperature.

CTX-M-14 was expressed and purified as described previously[73]. The pCR4:CTX-M-14 plasmid was transformed into E. coli strain BL21 (DE3) and grown in LB medium supplemented with 100 µg/mL ampicillin at 37 °C until an OD$_{600}$ of 0.6–0.7 was reached. Protein expression was induced by addition of IPTG to a final concentration of 150 µM and the cells were further incubated at 37 °C for 4 h. The cells were harvested by centrifugation (5500 × g, 10 min, 4 °C) and pellets were stored at −20 °C until purification. The pellets were resuspended in purification buffer (20 mM MES, pH 6) and sonicated for lysis. Cell debris was separated by centrifugation (20,000 × g, 1 h,

4 °C) and the supernatant was dialyzed overnight against a large volume of purification buffer at 4 °C using a 6–8 kDa molecular weight cut-off membrane. The CTX-M-14 protein was purified using cation exchange chromatography (5 ml HiTrap SP FF, Cytiva) and eluted using a gradient of 20 mM MES, pH 6, 0–50 mM NaCl over 5 column volumes. For crystallization, the protein was then concentrated to 22 mg/ml using 10 kDa centrifugal filter units (Amicon Ultra-15). Microcrystals were obtained in a batch crystallization approach, mixing 50% (v/v) purified protein with 45% (v/v) crystallising agent (40% (w/v) PEG 8000, 200 mM LiSO$_4$, 100 mM sodium acetate, pH 4.5) and with 5% (v/v) undiluted seed stock solution. This resulted in crystals with a homogeneous size distribution of 11–15 µm after approximately 90 min. To stop further crystal growth crystals were centrifuged at 200 × g for 5 min and the supernatant was replaced with a stabilisation buffer (28% (w/v) PEG 8000, 140 mM LiSO$_4$, 70 mM sodium acetate, 6 mM MES pH 4.5, 15 mM NaCl). For droplet injection 0.333 M piperacillin solution was freshly prepared in reaction initiation buffer (140 mM LiSO$_4$, 70 mM sodium acetate, 6 mM MES pH 4.5) prior to data collection and stored at room temperature.

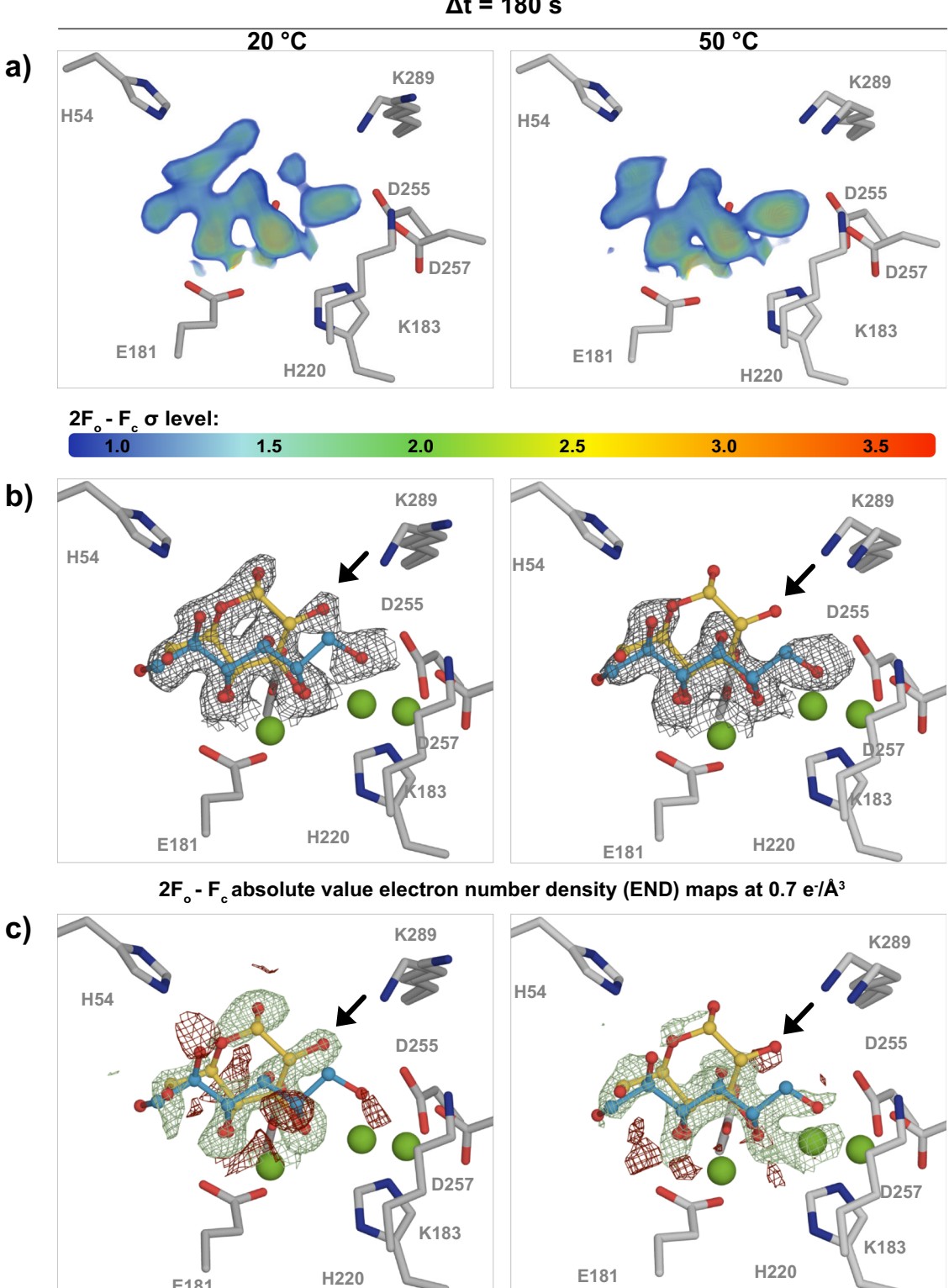

**Fig. 5 | Increased temperature reveals reaction intermediates.** XI active site comparison, 180 s after reaction initiation at 20 °C and 50 °C; **a** $2F_o - F_c$ density shown at RMSD levels from 1.0−3.5, strongest density is shown in red; **b** $2F_o - F_c$ absolute value electron number (END) maps shown at 0.7 $e^-$ / $Å^3$; **c** $F_o - F_o$ difference electron density (DED) maps shown at 2.7 ± RMS. An arrow highlights the electron density change around the closed-ring glucose O2-oxygen. The comparison at an increased delay time demonstrates that certain reaction intermediates are only available via an increased temperature, and cannot be resolved by increasing the delay time.

**Table 1 | Nano-DSF buffer for $T_M$ determination**

| | |
|---|---|
| CTX-M-14-buffer 1 | 20 mM MES pH 6.0, 50 mM NaCl |
| CTX-M-14-buffer 2 | 140 mM $Li_2SO_4$, 15 mM NaCl, 70 mM NaOAc pH 4.5, 6 mM MES, 28% PEG 8000 |
| CTX-M-14-buffer 3 | 140 mM $Li_2SO_4$, 15 mM NaCl, 70 mM NaOAc pH 4.5, 6 mM MES |
| CTX-M-14-buffer 4 | 140 mM $Li_2SO_4$, 15 mM NaCl, 70 mM NaOAc pH 4.5, 6 mM MES, 1.5% PEG 8000 |
| XI-buffer 1 | 20 mM HEPES pH 7.5 |
| XI-buffer 2 | 10 mM HEPES pH 7.5, 0.2 M $Li_2SO_4$, 25% PEG 3350 |
| XI-buffer 3 | 10 mM HEPES pH 7.5, 0.2 M $Li_2SO_4$ |
| XI-buffer 4 | 10 mM HEPES pH 7.5, 0.05 M $MgCl_2$ |

### Determination of the melting temperatures

The melting temperatures of CTX-M-14 and XI were determined in four different buffer systems each (Table 1). The buffers tested were the protein storage buffer (buffer 1), crystallization buffer (buffer 2), crystallization buffer without PEG (buffer 3) and activity assay buffer (buffer 4). The proteins were diluted to 1 mg/mL in the respective nanoDSF buffers and incubated for 20 min on ice. Standard grade nanoDSF capillaries (Nanotemper) were loaded into a Prometheus Panta (Nanotemper), excitation power was adjusted to 25% and samples were heated from 20 °C to 95 °C with a slope of 1 °C/min. All samples were examined in triplicates and error bars represent standard deviations.

### Calculation of $C_\alpha$-RMSD

In order to quantify and visualise the changing global structural differences, a pairwise backbone Root Mean Square Deviation (C$\alpha$ RMSD) was calculated for both model systems and represented as a categorical heatmap (Supplementary Fig. 1). Before calculating the RMSD for a pair, the two structures were aligned via Singular Value Decomposition (SVD) using the Biopython package `Bio.SVDSuperimposer` in order to minimise the resulting RMSD value. A temperature dependent clustering is observed for both model systems.

### Beamline experimental setup and X-ray data collection

Diffraction data were collected at the EMBL endstation P14.2 (T-REXX) at the PETRA-III synchrotron (DESY, Hamburg) with an X-ray beam of $10 \times 7$ µm (H × V) on an Eiger 4M detector (Dectris, Baden-Daettwil, Switzerland). Data collection was conducted as previously described[40] within the environmental control box mounted on the T-REXX endstation. Briefly: microcrystals mounted in a HARE-chip solid target containing 20,736 wells were moved through the X-ray beam using a 3-axis piezo translation stage setup (SmarAct, Oldenburg, Germany)[63]. Time delays were generated via the HARE method and reaction initiation was achieved via in-situ droplet injection via the LAMA method, as described in detail below[40,43,63]. In brief, the HARE method reduces total wall-clock time for long time-delays, while the LAMA method is a means of reaction initiation via in-situ droplet deposition. As a broad guideline approximately 5000 still diffraction patterns were recorded per time-point as previously determined to be sufficient[42].

### Reaction initiation via the LAMA method

Combination of our fixed-target SSX approach[40,63] with diffusion based reaction initiation was enabled via a piezo-driven droplet injector. The method is called 'liquid application method for time-resolved analyses' (LAMA)[43]. To this end, protein micro-crystals were mounted into the features of the chip via a gentle vacuum leaving only residual amounts of mother liquor covering the crystals. As a requirement for efficient reaction initiation, highly concentrated ligand solutions (see above) were freshly prepared, sterile filtered, and degassed directly prior to the experiment. The ejection capillary (LAMA nozzle) was placed at a

distance of ca. 1 mm in front of the HARE chip and carefully aligned to the crystal wells making use of the infrared light on-axis-viewing system. Both the humidity control stream described in the original LAMA paper, as well as the Mylar sheets, which cover the chip surfaces to prevent crystal dehydration, have become obsolete in the environmental control box and were therefore omitted during these experiments. Generally, in the LAMA approach, single droplets with a volume of 75–150 pl are shot onto each of the chip-mounted protein microcrystals, resulting in a total ligand consumption of approximately 1.5 µl per chip with 20,736 features. Briefly, a reaction initiation cycle proceeds as follows: (1) the chip moves into position, (2) the detector is triggered, and a $t_0$ image is recorded with an exposure time of 5 ms; (3) a TTL pulse triggers the droplet injector. One droplet is ejected at a velocity of approximately 2 m/s, that is a travel time of ca. 0.5 ms before it reaches the protein micro-crystal, which is taken into consideration during the post-exposure time (5 ms). (4) The crystals are exposed to X-rays and a $t_1$ image is recorded after the specified delay time.

### Data processing and structure determination

Diffraction data were processed using the CrystFEL v0.10.2 package[74]. After an initial solution from PHASER[75] using 6RNF, 6GTH, all further models were refined from this initial structure against the individual temperature-dependent datasets and 5D-SSX data using phenix.refine. *phenix.refine*. Iterative cycles of refinement and manual model building of additional and disordered residues in COOT-v0.8 were used to address the specific changes in each structure[76–78]. Occupancy refinement was carried out in *phenix.refine* by defining constrained occupancy groups of ligand-free and ligand-bound proteins, and refining against all states simultaneously. Further details are given in the supplementary notes.

**Electron density figures.** Molecular images were generated in PyMOL[79]. The $2F_o - F_c$ electron density as volume elements with different RMSD levels were generated in PyMol *via* the volume command (`volume volumename, mapname, level, ligandname, carve=1.4`), where colour and RMSD-level settings according to Supplementary Table 10 have been used. $2F_o - F_c$ absolute value electron number (END) maps were calculated using the END/RAPID script as previously described[80]. $F_o - F_o$ difference electron density (DED) maps were calculated using the MatchMaps software (version 0.6.6), subtracting the apo dataset at a particular temperature from the reaction-triggered delay-time dataset at the respective temperature according to default parameters[81]. Ligand occupancy was displayed via the following command: `spectrum q, teal_hotpink, imp, minimum=0.1, maximum=0.6` for CTX-M-14 and `spectrum q, teal_hotpink, imp, minimum=0.3, maximum=0.5`. POLDER-OMIT maps were generated using *phenix.polder*, omitting the ligand residues in the constrained occupancy groups[82].

**Fitting ADPs to a shifted inverse gamma distribution.** The isotropic ADP frequency histograms of each structure were fitted with a three parameter Shifted Inverse Gamma Distribution (SIGD) function as shown previously by Masmaliyeva et al.[64]. However, in contrast to the maximum-likelihood estimation using the Fisher scoring method applied previously[64], parameter estimation and optimization were carried out by using non-linear least squares with the Python module `scipy.optimize.curve_fit`. To produce reasonable estimates, necessary parameter restraints[64] were applied during the estimation process. Briefly, the starting value of the shift parameter was taken to be equal to 90% of the minimum of the ADPs in the PDB file.

**Sub-dataset splitting and analysis with RoPE.** For the RoPE analysis, each dataset was split into subsets of at least 2000 diffraction patterns each using *partialator*[74], which were then independently refined with DIMPLE[83], without human intervention. This allows us to assess the

relative contributions of random fluctuation and genuinely temperature-dependent changes within the crystal structure, as employed previously[84]. The output of these PDBs was analysed in RoPE[66] the top three coordinates were rotated for clarity and projected onto 2D of the atomic coordinate differences, and corresponding temperature metadata.

## Reporting summary

Further information on research design is available in the Nature Portfolio Reporting Summary linked to this article.

## Data availability

All crystallographic data have been deposited in the protein data bank under https://www.rcsb.org/. Data for the datasets of XI and CTXM-14, respectively, have been deposited under the accession numbers: 9G5N, 9G5S, 9G5W, 9G5X, 9G61, 9G6L, 9G6M, 9G6N, 9G6O, 9G6P, 9I79, 9I7L, 9G7V, 9G7W, 9G7X, 9G7Y, 9G7Z, 9G80, 9G81, 9G82. Further details are available in supplementary tables 6-10. The previoulsy published structures can be access under following accession codes: 3Q1F, and 3KCL, respectively. All information that is required for reproduction of the hardware, including 3D-PDFs, CAD files, step-files, electronics, and a bill of materials can be obtained under the following https://doi.org/10.5281/zenodo.12758835. Source data are provided as a Source Data file. Source data are provided with this paper.

## Code availability

All custom code is available in the Supplementary code section and via https://doi.org/10.5281/zenodo.12758835.

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

## Acknowledgements

All multi-temperature SSX data were collected at endstation P14.2 (T-REXX) operated by EMBL Hamburg at the PETRA-III storage ring (DESY, Hamburg, Germany). We acknowledge technical support by the SPC facility at EMBL Hamburg. We would like to thank our colleagues G. Bourenkov, M. Agthe, J. Holton, and A.R. Pearson for their continuous support, helpful discussions or critical reading of the manuscript. Construction of T-REXX was funded by the BMBF (Verbundforschungsprojekt 05K16GU1, 05K19GU1, and 05K22GU6). T-REXX beamtime was awarded as part of the EMBL BAG MX-660 and MX-1008. The authors gratefully acknowledge the support provided by the Max Planck Society and the Cluster of Excellence 'The Hamburg Centre for Ultrafast Imaging' of the Deutsche Forschungsgemeinschaft (DFG) (EXC 1074, project ID 194651731) and the Joachim Herz foundation (Biomedical physics of infection) (E.C.S.) and from the Joachim Herz Stiftung add-on fellowship (P.M.). Additional funding was provided by the DFG *via* grant No. 451079909 to P.M. E.C.S. acknowledges support by the DFG via grant no. 458246365 and by the Federal Ministry of Education and Research, Germany, under grant number 01KI2114. Funded by the European Union (ERC, DynaPLIX, SyG-2022 101071843). Views and opinions expressed are, however, those of the authors only and do not necessarily reflect those of the European Union or the European Research Council. Neither the European Union nor the granting authority can be held responsible for them.

## Author contributions

P.M. and E.C.S. designed the experiments; E.C.S. and P.M. performed the experiments with support from D.v.s., A.P. and F.T.; P.M., A.P., C.E.H. and K.B. prepared protein and the protein crystals; E.C.S., P.M., J.P.L. and F.T. designed and built the environmental control box; D.v.s., A.P., E.C.S. and P.M. processed and analysed the diffraction data; P.M., E.C.S., H.S., J.P.L. and F.T. characterized the control box; G.G. performed the SIGD analysis. H.M.G. performed the RoPE analysis. F.T. has made the numerical calculations. P.M. and E.C.S. wrote the manuscript; All authors discussed and corrected the manuscript.

## Funding

## Competing interests

The authors declare no competing interests.
