## [Transparent Peer Review file · Nature Communications]

Probing the modulation of enzyme kinetics by multi-temperature, time-resolved serial crystallography

Corresponding Author: Dr Eike Schulz

Version 0:

Reviewer comments:

Reviewer #1

(Remarks to the Author)

The authors describe methods and results of changing the temperature in time-resolved crystallographic experiments on enzymatically catalyzed, irreversible reactions. This will enable the introduction of temperature to structure based enzymology with the goals to determine barriers of activation and to observe temperature induced structural changes in working enzymes. The work is well supported by experimental evidence. Results are well described and can be reproduced if necessary. The presented results pave the way for routinely changing a critical variable, temperature (in addition to humidity), in time-resolved crystallographic experiments at synchrotrons and Xray free electron laser light sources using fixed targets.

The authors should expand on RoPe. Describe the RoPe space (e.g. what are the basis vectors of the RoPe space) and give an intuitive description of the RoPe components (or modes). Is there a dependence of the RoPe space on the number of DPs. Are 2000 DPs sufficient for reliable DIMPLE refinement? This is particularly important since the authors state "As a broad guideline approximately 5000 still diffraction patterns were recorded per time-point as previously determined to be sufficient".

L663: represented as a categorical heatmap (Sup. Fig. ??).
Please provide the double questionmarked information.

Material and Methods:

Please specify whether the models PDB-ID: 6RNF, 6GTH where indeed run through phaser for a full molecular replacement or did they already constitute good enough initial models for refinement.

Sup Fig. 5:

There are no diffraction patterns shown, please reformulate.

The first three lines of the python code are interleaved with Sup. Tab. 6, please disentangle.

Reviewer #2

(Remarks to the Author)

Report on

Probing the modulation of enzyme kinetics by multi-temperature, time-resolved serial crystallography, Eike C. Schulz, Andreas Prester, David von Stetten, Gargi Gore, Caitlin E. Hatton, Kim Bartels, Jan-Philipp Leimkohl, Hendrik Schikora, Helen M. Ginn, Friedjof Tellkamp, Pedram Mehrabi.

This article describes a very nice technical innovation which is likely to be of value to the time-resolved X-ray diffraction community. The article motivates well the benefits of combining temperature as a fifth parameter, along with time-dependent

atomic coordinates. The introduction sets the scene well and makes appropriate reference to the scientific literature. In particular, the somewhat similar approach of Marius Schmidt [Schmidt et al., Five-dimensional crystallography. *Acta Crystallogr. A*, 66, 198–206 (2010); Schmidt, M. et al. Protein energy landscapes determined by five-dimensional crystallography. *Acta Crystallogr. D*, 2534–2542 (2013)] and more recent work by James Fraser, Michael Thompson et al... [Wolff, A. M. et al. Mapping protein dynamics at high spatial resolution with temperature-jump x-ray crystallography. *Nature chemistry* 15, 1549–1558 (2023)] are cited very positively, as is appropriate.

The weakness of the article is that, having built up a strong scientific case for the new insights that are likely to emerge from high-quality five-dimensional X-ray crystallography data, it somewhat falls flat since there is no core scientific message from the crystallographic results. I trust the structural analysis of the authors and I really appreciate the effort that they have made to build up this instrument, but either the scientific message is lacking or the communication of the scientific message is understating the findings. Thus, while I absolutely believe that the work deserves to be published, it is not obvious to me why the incorporation of serial crystallography data into the five-dimensional crystallography approach of Schmidt et al. more than a decade ago, would warrant the additional impact of publishing in *Nature Communications*. Thus, I recommend that the authors be invited to revise the article to highlight the scientific value of the new structural insights to emerge from their data. If they cannot do this, then perhaps a more specialized journal is appropriate.

Specific comments on the manuscript:

On the whole the paper is well written, and I have confidence the crystallographic analysis of the authors. Moreover, their technical innovations are well described. Therefore, I am focusing the remainder of this report on the core issue with the manuscript, which is the need to improve the communication of their scientific results.

The authors' choice of temperature domain from – 10 °C to 70 °C is appropriate for the goals of the method.

Introduction:

A few words explaining the enzymatic function of both enzymes of choice (extended spectrum β -lactamase and xylose isomerase) seem to be appropriate in the introduction.

The introduction makes the rather bold claim that “For both systems we show that a modification of the environmental temperature modulates the underlying structural dynamics as well as the enzyme kinetics, enabling to resolve alternate structural intermediates not observed at high occupancy at ambient temperatures.” I think that the extent to which this goal is met gives a fair basis whether or not to recommend publication in *Nature Communications*.

Results and discussion:

The major technical innovation of the work, the environmental control box design, is well described and very nice. The authors deserve credit for this innovation.

Figure 1:

An important conclusion is that CTX-M-14 undergo little progression in the B-factor distribution (atomic displacement parameters) from 10 °C to 40 °C and then undergo a sudden jump at 50 °C, because it has a melting temperature of 53 °C (as measured using differential scanning interferometry). By contrast, Xylose isomerase shows a steady progression in the evolution of B-factors over the entire temperature domain, but does not reach its melting temperature of 83 °C.

Figure 2:

I'm afraid that I do not understand exactly what is being presented in Figure 2a and 2b. Please explain RoPE space, and what the two principle components being plotted in the graph mean physically. There appears to be some temperature dependent correlation, but it is impossible to judge the significance of this.

Figure 2c and 2d are much clearer. This representation establishes that some loop regions show larger angular distributions (ie. become more disordered) with higher temperature. This is very nice to measure directly, but seems to fall a little short of the claims of the introduction.

In the time-resolved studies, the authors build upon their own earlier work and initiate reactions using the LAMA-method. But this method is not explained in the document, rather the authors refer to earlier publications. Since the method of reaction initiation is so central to the major findings of time-resolved crystallography studies, it seems appropriate (or essential) to also describe the method of reaction initiation in this work, even if it does overlap with earlier publications. Please also state the time-scale to which this reaction initiation method is synchronized.

Figure 3:

This shows electron density ensembles of structures 3 seconds after reaction initiation for extended spectrum β -lactamase and 60 seconds after reaction initiation for xylose isomerase. This is clearly the key figure against which the authors' claims that “For both systems we show that a modification of the environmental temperature modulates the underlying structural dynamics as well as the enzyme kinetics, enabling to resolve alternate structural intermediates not observed at high occupancy at ambient temperatures.”

Unfortunately, the authors have tried to be too sophisticated in their presentation of panels 3a and 3d, to the extent that these become impossible to read and for temperature dependent distinctions to be made. They are trying to represent the $2F_{obs}-F_{calc}$ map from sigma levels from 1.0 sigma to 3.5 sigma in the same figure and it doesn't work. Both need to be completely redrawn, as in panels 3c and 3f, which are interpretable. This failure to communicate a key figure is a major shortcoming of the work. As was shown half a century ago [R. Henderson, J. K. Moffat, The difference Fourier technique in protein crystallography: Errors and their treatment. *Acta Crystallogr. B* 27, 1414–1420 (1971)], difference Fourier electron density maps are much more sensitive than $2F_o-F_c$ maps. If there are real differences in the observations, they should be clear from difference Fourier maps. The superposition of structures does not tell me anything I can interpret in any meaningful way. The Polder omit maps (Figure 3c and 3f) are better, but it is difficult to distinguish the Polder omit map electron density reported for xylose isomerase at all three temperatures, except for the possibility that the ring-structure has slightly lower occupancy for higher temperature. The Polder omit map electron density of extended spectrum β -lactamase does show much lower occupancy at 20 °C than at higher temperature.

Thus, I am not convinced that the authors achieve the stated goal. Moreover, these electron density differences do indicate that the reaction proceeds faster at higher temperature in both cases, but this is short of the claim of “enabling to resolve alternate structural intermediates not observed at high occupancy at ambient temperatures”. In both cases, the authors acknowledge: “The multiple overlapping states make an unambiguous interpretation based on the electron density alone difficult”. While this is true, and the very impressive method developed is pointing towards a solution, I don't yet see clear evidence that this problem has been overcome with this method.

Methods:

On the whole the methods are clear and report the approach to a level appropriate for scientific publication. However, the LAMA reaction initiation approach is not described in any detail, and thus the reader must access other articles to understand what was done. Moreover, the choice of only one time-delay (3 sec in one case, 60 sec in another) has not been motivated in the article.

Line 663 refers to Sup. Fig. ???. Please correct.

Summary:

This is a really nice technical innovation which is well designed and well presented. I think that the work is truly very nice and deserves publication. However, in presenting their case, the authors set the bar that the method is providing genuine new structural insight which would not be available using other classical approaches (working at lower temperature but waiting for longer time-delays for example). It is possible that a restructuring of the document to better emphasise that this goal has been achieved may be possible, especially by redrawing Figure 3, the critical figure, so as to communicate much better the key findings with clarity. This would include the presentation of difference Fourier analysis, and showing $2F_{obs}-F_{calc}$ maps with only one contour level. I would like to give the authors a chance to try to do this since the work, overall, is very impressive. But if they cannot do this, then perhaps another journal is a better option?

Reviewer #3

(Remarks to the Author)

This manuscript describes the temperature dependence of time resolved crystallography. Unfortunately, the temperature is simply taken as the value that the sample was equilibrated to. The actual temperature is not known and it is not addressed in this manuscript. I cannot support publication of this manuscript.

Version 1:

Reviewer comments:

Reviewer #1

(Remarks to the Author)

My concerns are sufficiently addressed.

Reviewer #2

(Remarks to the Author)

Report on

Probing the modulation of enzyme kinetics by multi-temperature, time-resolved serial crystallography, Eike C. Schulz, Andreas Prester, David von Stetten, Gargi Gore, Caitlin E. Hatton, Kim Bartels, Jan-Philipp Leimkohl, Hendrik Schikora, Helen M. Ginn, Friedjof Tellkamp, Pedram Mehrabi.

This is a revised version of this manuscript. The authors have made sincere and respectful efforts to address the comments

of reviewers in a constructive way. As I wrote in my previous report, the article motivates well the benefits of combining temperature as a fifth parameter, along with time-dependent electron density and refined atomic coordinates. In revising their article, the authors include more background to the biological systems of study, and have included data from longer time-delays to make the case that temperature is critical to optimize the occupancy of transient reaction conformations of interest. These are additions that improve the main message of the paper, as well as the accessibility of the work to the general reader. I am impressed by the technical aspects of the work overall. I am confident that the changes made to the article, and the argument made by the authors concerning the potential impact of the methods they have developed on the field as a whole, warrant publication in a broad audience journal. I therefore recommend publication of this version of the article in Nature Communications.

May I also take the opportunity to apologize to the authors for the delay in my report. The request came in during an extended period of exceptional time pressure, and I completed this report as soon as this had passed.

Reviewer #3

(Remarks to the Author)

The rebuttal does not address the concern. The temperature is not known. I cannot recommend publication of this manuscript.

Reviewer #4

(Remarks to the Author)

The manuscript by Schulz, et al. describes the implementation of "5-dimensional" macromolecular crystallography at a synchrotron X-ray source. I note that the editor of the journal specifically requested that I provide my opinion on the accuracy of the temperature control in the reported experiments, especially with respect to X-ray beam-induced heating. Therefore, as requested, I will limit my comments to focus on this aspect of the paper exclusively.

I do think that reviewer #3 is correct that there will be some beam-induced X-ray heating of the sample. However, I strongly disagree with the notion that this is a reason to not publish the manuscript. It would be excellent to know the instantaneous temperature of the sample during the measurement; however, I think the 5D crystallography experiment described by the authors is still valuable in the absence of accurate temperature determination.

The 2024 paper by Baxter, et al. (PMID: 38848551) demonstrates that there is significant X-ray beam induced heating in serial synchrotron crystallography (SSX), which is almost certainly similar in the reported 5D experiments. In fact, in the 2024 paper, Baxter and colleagues provide a relationship between the X-ray dose and beam induced heating. It seems that this relationship could also be applied by the authors of the manuscript under review to estimate the temperature of the samples during the measurement. In the manuscript, the X-ray dose per shot is not given, so I cannot estimate myself. In this regard, I agree with reviewer #3.

On the other hand, I think it is important to bear in mind the real utility of multi-temperature crystallography, which is to alter the populations of conformational states in the crystal, so that one can observe a richer picture of the conformational landscape. The B-factor plots shown in Figures 1b and 1c, representing a shift toward more disorder at higher temperatures, illustrate that this is indeed happening in the context of the reported experiments. Therefore, the time-resolved measurements at different temperatures could reveal different substates of the molecule that are accessible during different stages of the catalytic cycle, and potentially reveal their couplings as with static, multi-temperature experiments. In my opinion, this is what makes the 5D approach promising. It is true that without knowing the exact temperature of the sample, one cannot do, e.g. an Eyring analysis, but it has also been demonstrated that reaction rates in crystallo and in solution often do not match, and therefore it is not useful to do this analysis from time-resolved crystallography experiments anyway.

Finally, I note that it is possible to measure the temperature directly from the water ring present in the X-ray diffraction images. There is substantial literature describing this in the liquid X-ray scattering field.

Version 2:

Reviewer comments:

Reviewer #4

(Remarks to the Author)

I am satisfied that the revised version of the manuscript and supplementary information adequately address the issues related to X-ray heating. I support the publication of this revised manuscript in its current form.

REVIEWER COMMENTS

We have left the reviewer comments in black sans-serif font. Please find our response to the continuously numbered reviewer remarks in blue serif-type font.

Reviewer #1 (Remarks to the Author):

The authors describe methods and results of changing the temperature in time-resolved crystallographic experiments on enzymatically catalyzed, irreversible reactions. This will enable the introduction of temperature to structure based enzymology with the goals to determine barriers of activation and to observe temperature induced structural changes in working enzymes. The work is well supported by experimental evidence. Results are well described and can be reproduced if necessary. The presented results pave the way for routinely changing a critical variable, temperature (in addition to humidity), in time-resolved crystallographic experiments at synchrotrons and X-ray free electron laser light sources using fixed targets.

#1

The authors should expand on RoPe. Describe the RoPe space (e.g. what are the basis vectors of the RoPe space) and give an intuitive description of the RoPe components (or modes). Is there a dependence of the RoPe space on the number of DPs. Are 2000 DPs sufficient for reliable DIMPLE refinement? This is particularly important since the authors state “As a broad guideline approximately 5000 still diffraction patterns were recorded per time-point as previously determined to be sufficient”.

- We have taken the opportunity to explain the plots in more detail within the main text:

“Briefly, the software RoPE begins by enforcing canonical bond length and angle geometry on every structure, and optimising torsion angles to account for this [citation of original algorithm]. For each bond, the mean torsion angle is calculated from all structures. For each structure, a list of torsion angle deviations from the mean is compiled. For every pair of structures, a correlation coefficient between these torsion angle deviations is generated. Singular value decomposition (SVD) is carried out on the full correlation matrix representing all structure pairs. RoPE then plots a RoPE space, transforming each structure into the coordinates represented by the U matrix. Three-dimensional plots represent the three highest ranked vectors of the subspace, rotated to maximise clarity, and projected onto 2D for the figure display. Therefore, these plots show the inherent variation between structural conformations as determined from their torsion angles.”

- One of the benefits of this style of analysis is demonstrated by considering the possible combinations of whether there is sufficient supporting data per structure, and whether there is the presence or absence of measurable signal. If the number of diffraction patterns is insufficient to support the analysis, then differences in torsion angles will be due to noise. Therefore, both a significant signal and a negligible signal will produce the same outcome, which is a lack of separative power in RoPE space. In the former case, this is a false negative. In the latter case it is an unsubstantiated negative. We include a table which summarises the behaviour of this analysis and it is hopefully clear that, due to a lack of a “false positive” outcome, the method is failsafe in the event that data is insufficient.

Outcome table for RoPE space

	Insufficient number of diffraction patterns	Sufficient number of diffraction patterns
Negligible signal	Unsubstantiated negative	True negative
Significant signal	False negative	True positive

- In addition we have extended the Supplementary methods section (see below) and provide an additional figure (Sup. Fig. 4) showcasing the titration of the number of diffraction patterns for RoPE analysis:

“RoPE space plots take advantage of the oversampling of serial diffraction data in order to establish a statistical support for interpretation of structural differences, by splitting the data for independent structure solutions. This needs to strike a balance, for a given number of diffraction patterns, between the total number of structures and the number of diffraction patterns assigned to each structure. The number of diffraction patterns will influence the quality of the data supporting the analysis. Here we vary the approximate number of diffraction patterns in each structure from 3000 to 8000 images, in steps of 1000 images. Naturally, with more images per structure, fewer structures are generated. We see that the separation is present for all image numbers per structure (Sup. Fig. 4 a), but to some extent, the “tightness” of each cluster increases with more supporting images per structure. Showing all structures at the same time, this time coloured by number of images (Sup. Fig. 4b), the clusters corresponding to respective temperatures are still recognisable. However, this shows that structures supported by fewer images are more similar to each other despite their different temperatures. This is likely because of poorer quality diffraction intensity estimates and greater reliance on the geometry term rather than the X-ray data during refinement. However, structures supported by more images can reveal more features associated with the given temperature, due to higher quality intensity estimates, and therefore are spaced further apart from one another.”

#2

L663: represented as a categorical heatmap (Sup. Fig. ??). Please provide the double question marked information.

- Corrected. Now pointing to Sup. Fig. 1

#3

Material and Methods:

Please specify whether the models PDB-ID: 6RNF, 6GTH were indeed run through phaser for a full molecular replacement or did they already constitute good enough initial models for refinement.

- The reviewer is correct. The structures were indeed good enough models for refinement. The section has been updated accordingly:

“[...] After an initial solution from PHASER using 6RNF, 6GTH, all further models were refined from this initial structure against the individual temperature-dependent datasets and 5D-SSX data using phenix.refine. Iterative cycles of refinement and manual model building of additional and disordered residues in COOT-v0.8 were used to address the specific changes in each structure. [...]”

#4

Sup Fig. 5:

There are no diffraction patterns shown, please reformulate.

- Indeed, the figure does not show diffraction patterns. However, the figure (now Sup. Fig. 9) shows whether diffraction patterns corresponding to a particular unit cell size have been recorded. We hope that the updated caption reduces the potential for confusion:

“Each feature on the HARE-chip that can contain a crystal is represented by a small grey square; if a diffraction pattern is recorded the particular square is highlighted in colour. A blue square indicates a diffraction pattern corresponding to a low humidity unit cell, while a red square corresponds to a diffraction pattern in the larger high-humidity unit cell. Humidity has been reduced by 5% for each row of compartments. a) recorded diffraction patterns corresponding to low-humidity unit-cells; b) recorded diffraction patterns corresponding to high-humidity unit-cells; c) overlay of low- and high-humidity unit-cell diffraction pattern hits; d) comparison of high- and low-humidity unit-cells.”

#5

The first three lines of the python code are interleaved with Sup. Tab. 6, please disentangle.

- Done.

Reviewer #2 (Remarks to the Author):

Report on

Probing the modulation of enzyme kinetics by multi-temperature, time-resolved serial crystallography, Eike C. Schulz, Andreas Prester, David von Stetten, Gargi Gore, Caitlin E. Hatton, Kim Bartels, Jan-Philipp Leimkohl, Hendrik Schikora, Helen M. Ginn, Friedjof Tellkamp, Pedram Mehrabi.

This article describes a very nice technical innovation which is likely to be of value to the time-resolved X-ray diffraction community. The article motivates well the benefits of combining temperature as a fifth parameter, along with time-dependent atomic coordinates. The introduction sets the scene well and makes appropriate reference to the scientific literature. In particular, the somewhat similar approach of Marius Schmidt [Schmidt et al., Five-dimensional crystallography. *Acta Crystallogr. A*, 66, 198–206 (2010); Schmidt, M. et al. Protein energy landscapes determined by five-dimensional crystallography. *Acta Crystallogr. D*, 2534–2542 (2013)] and more recent work by James Fraser, Michael Thompson et al... [Wolff, A. M. et al. Mapping protein dynamics at high spatial resolution with temperature-jump x-ray crystallography. *Nature chemistry* 15, 1549–1558 (2023)] are cited very positively, as is appropriate.

The weakness of the article is that, having built up a strong scientific case for the new insights that are likely to emerge from high-quality five-dimensional X-ray crystallography data, it somewhat falls flat since there is no core scientific message from the crystallographic results. I trust the structural analysis of the authors and I really appreciate the effort that they have made to build up this instrument, but either the scientific message is lacking or the communication of the scientific message is understating the findings. Thus, while I absolutely believe that the work deserves to be published, it is not obvious to me why the incorporation of serial crystallography data into the five-dimensional crystallography approach of Schmidt et al. more than a decade ago, would warrant the additional impact of publishing in *Nature Communications*. Thus, I recommend that the authors be invited to revise the article to highlight the scientific value of the new structural insights to emerge from their data. If they cannot do this, then perhaps a more specialized journal is appropriate.

- Undoubtedly, Schmidt et al. pioneered the field of 5D-SSX. However, in our opinion serial time-resolved crystallography holds many practical advantages over time-resolved crystallography using single crystals. Serial crystallography enables working on enzymes with irreversible reactions that cannot (at least not easily) be addressed via canonical single-crystal TRX. At the same time, serial crystallography minimizes the effects of radiation damage, which is a particular concern at ambient temperatures. Via the combination with our HARE technology this gains additional advantages as it enables addressing versatile time-delays up to the minutes domain. This is a critical advantage as the majority of enzymes have turnover times in the high ms time domain ($k_{cat} \sim 10/s$) (Bar-Even 2011, *Biochemistry*). Moreover, single crystal work is usually limited in the means of reaction initiation. In contrast to this, due to their small crystal size, versatile means of reaction initiation have been implemented for serial crystallography. Via our LAMA method that we combine with the multi-temperature work, we circumvent the need for optical triggering and rather enable reaction initiation via the most simple and most versatile way - that is in situ mixing. This enables countless systems to be addressed via time-resolved - and now also multi-temperature - crystallography. Finally, our new environmental control system also circumvents the need for sleeves or capillaries that are typically required for the humidity control of single crystals. While its importance is long known, our environmental control system also permits tailoring the humidity surrounding the

crystals thereby enabling to adjust system specific humidity levels. In summary, a unique strength of our approach is the combination of all of the advantages stated above. This will finally make the original 5D-SSX approach - only accessible to the true expert - available to the growing community of macromolecular crystallographers and structural biologists, who seek detailed insight into catalytic mechanisms via time-resolved crystallography. The versatility and general applicability of the method described in this manuscript, now enables this large group of researchers to push the boundaries of what is experimentally possible and in our opinion therefore warrants the publication in a general interest journal. Finally, we clearly demonstrate new results that critically depend on the implementation of 5D-SSX, namely demonstrating in a structural context that reaction turnover is directly linked to temperature, and that reaction key intermediates can only be determined at these elevated temperatures (Fig. 4).

- To clarify the advantages of combining serial time-resolved with multi-temperature crystallography we have modified and extended the introduction as follows:

“[...] Here we present a novel method that enables multi-temperature, time-resolved serial crystallography (5D-SSX) experiments. Our primary objective is to specifically address the interplay between protein structure, dynamics and activity as a function of temperature. By maintaining the protein crystals at a defined temperature and humidity, our method enables studying molecular motions in response to physiological events, especially at comparably long time-delays, which are often encountered during enzyme catalysis [33]. Especially the effects of relative humidity on crystals’ unit cell have been noted early on by Perutz and Kendrew, who noted swelling or shrinkage of hemoglobin crystals [34, 35]. This was later used to tailor the diffraction properties of single protein crystals [36–39]. Via our environmental control box this is now accessible to serial crystallography, too. The implementation of our hit-and-return (HARE) serial crystallography approach permits to conveniently address irreversible enzymatic reactions that escape time-resolved analyses within single macroscopic crystals [40]. Moreover, employing a serial crystallographic approach also has clear advantages with respect to radiation damage, as the total dose is distributed over several thousand individual crystals [41, 42]. Additionally, we utilise our liquid-application method for time-resolved analyses (LAMA) [43]. This enables to also address the predominant fraction of enzymes that are not natively photoactive [44], but rather using the most simple way of reaction initiation by directly mixing soluble ligands with the protein crystals [45]. These features render our method particularly versatile and enable addressing a wide spectrum of different systems. To demonstrate the versatility of the method, we have employed two different model systems: a mesophilic enzyme[...]”

Specific comments on the manuscript:

On the whole the paper is well written, and I have confidence the crystallographic analysis of the authors. Moreover, their technical innovations are well described. Therefore, I am focusing the remainder of this report on the core issue with the manuscript, which is the need to improve the communication of their scientific results.

The authors’ choice of temperature domain from – 10 °C to 70 °C is appropriate for the goals of the method.

Introduction:

#6

A few words explaining the enzymatic function of both enzymes of choice (extended spectrum β -lactamase and xylose isomerase) seem to be appropriate in the introduction.

- To address the reviewer's comment regarding the enzymatic function of our two model systems we have extended the introduction as follows.

“[...] Extended spectrum, class-A serine beta-lactamases (ESBL, CTX-M, EC:3.5.2.6) have been identified in a growing number of clinical isolates of Gram-negative bacteria, with worldwide distribution. Klebsiella pneumoniae CTX-M-14 is a monomeric enzyme that catalyzes the hydrolysis of penicillin-, cephem- and carbapenem-family antibiotics. Canonically, a carboxylate or negatively charged group on the beta-lactam antibiotic binds to positively charged residues in the active site. This is followed by acylation of the beta-lactam ring, which subsequently forms a covalent bond with the catalytic serine residue. Subsequent to this, an activated water molecule hydrolyses the acyl-enzyme intermediate, leading to the release of the ring-opened, inactivated antibiotic. Previous reports on in-solution kinetics vary considerably depending on substrate and reaction conditions. For instance, the turnover constants (k_{cat}) for CTX-M-14 with cefotaxime demonstrated k_{cat} values between 37/s – 1400/s. This prompted us to test a variety of alternative substrates, from which we selected piperacillin as the model compound for this study.

Streptomyces rubiginosus xylose isomerase (XI, EC:5.3.1.5), is a tetrameric enzyme that facilitates the interconversion of aldoses to ketoses, for example D-xylose into D-xylulose or D-glucose into D-fructose. Its catalytic reaction depends on two divalent metal ions and a histidine residue in the active site. The initial formation of an open chain intermediate is followed by its conversion into a 5-membered ketose ring via a hydride shift mechanism. The in-solution k_{cat} of XI has previously been determined to be 53/s at an optimal reaction temperature of approximately 80 °C (pH 7.5). In contrast, at 60 °C (pH 7.3) the k_{cat} for glucose has been reported to be approximately 5.3/s, while at 35 °C, XI has been reported to be nearly inactive.[...]”

#7

The introduction makes the rather bold claim that “For both systems we show that a modification of the environmental temperature modulates the underlying structural dynamics as well as the enzyme kinetics, enabling to resolve alternate structural intermediates not observed at high occupancy at ambient temperatures.” I think that the extent to which this goal is met gives a fair basis whether or not to recommend publication in Nature Communications.

- For both systems we have shown a response of the underlying structural dynamics via our SIGD-analyses that show a global change in ADPs as a function of temperature (Fig. 1), as well as via altered torsion angle dynamics (Fig. 2). The altered kinetics are addressed in Figs. 3-5). As the reviewer acknowledges, every electron density in a time-resolved experiment is an ensemble of superimposed substates with respective fractional occupancies that contribute to the same electron density. Often this is the source of ambiguity that makes a clear interpretation and hence an assignment of specific intermediates difficult, if not impossible. In turn, it follows that any form of enrichment of a particular sub-state that leads to a cleaner electron density with

better interpretability will largely improve this situation. While our data cannot cover any eventuality they clearly support the above mentioned claims:

1. In the case of CTX-M-14, a new conformation of the product complex could be observed within the wildtype protein, that has not been observed previously and would likely not be observable via conventional mutational methods, due to the underlying steric repulsion. However, more importantly these data show that by a temperature modification at the same time-delay (3 s) we were able to enrich particular reaction intermediates, that is either the covalent intermediate at 30 °C or the product complex at 37 °C. In either case, the electron density still displays the aforementioned ensemble of states, but the ability to modify both time and temperature holds the potential to sufficiently enrich a particular sub-state for unambiguous identification. To the best of our knowledge, this has not been demonstrated for an enzyme catalyzing an irreversible reaction before.
 2. In the case of XI, we observed a complex with an open-ring glucose intermediate in the wild-type protein that has not been observed before. To demonstrate that particular intermediates can be sufficiently enriched, or only observed at a particular temperature, we carried out a temperature-resolved comparison at an extended time delay (180 s) (Fig. 5). This clearly demonstrates that at 20 °C there is almost no change in the closed-ring structure, while at 50 °C, the majority of the density can be explained by an opening intermediate. These data unambiguously demonstrate that structural intermediates can be resolved that are not visible (at high occupancy) at ambient temperatures.
- We have modified the text accordingly:

“For CTX-M-14 the 20 °C structure corresponds to the unbound state, exhibited by rather discontinuous difference electron density and correspondingly low occupancy for any of the piperacillin intermediates. The 30 °C structure on the other hand is clearly populated by difference density, corresponding to ligand molecules. Occupancy refinement indicated a mixture corresponding predominantly to the acyl-enzyme-intermediate and a complex with the hydrolysed piperacillin product. Finally, the 37 °C structure corresponds predominantly to CTX-M-14 in complex with the hydrolysed piperacillin product (Fig. 3 a-c). These different catalytic states are further supported by alternate conformations of the catalytic serine (S70) that shifts its position during catalysis, as indicated by an altered population of the difference electron density (Fig. 3 d). Interestingly, the conformation of the piperacillin hydrolysis product is clearly distinct from a previously reported complex in a mutant enzyme (Sup. Fig.3) While the mutant structure (3QIF) adopts a conformation akin to the acyl-enzyme intermediate, our product complex obtained in the wild-type protein and at physiological temperature is shifted between 1.8 Å and 3 Å from that conformation. Considering the overarching clinical relevance of ESBLs for bacterial infections, this novel product complex conformation emphasises the importance of detailed structural information under physiological conditions.

Since different catalytic sub-states of CTX-M-14 can clearly be distinguished at different temperatures (unbound state, acyl-enzyme intermediate, hydrolysed piperacillin product (Fig. 3 a-c)), these data clearly show that both ligand diffusion and turnover kinetics of mesophilic enzymes can be modulated by temperature variation in our control box. Temperature modification can therefore become an important tool for the selective enrichment and differentiation of specific metastable or stable reaction intermediates in the context of time-resolved serial crystallography.”

“[...] However, as can be seen from the END and DED map analysis, different catalytic states (i.e. closed- and open-chain intermediates) can be detected at the same delay time as the temperature is increased. This shows that there is a direct correlation between the increase in the catalytic activity of XI and the rise in temperature. This observation indicates that thermal modulation can be utilised as a method to derive shifts in reaction equilibria, thus facilitating the identification of intricate mechanistic distinctions that depend on the energy of the system [...].”

“[...] Next, we addressed the question of whether temperature is a critical parameter for the observation of certain reaction intermediates or whether this can also be compensated for by longer delay times at lower temperatures. To this end, we compared the XI active centre at a delay time of 180 s at 20 °C or 50 °C (Fig. 5). In practice, this delay time should be long enough to accumulate reaction intermediates even for enzymes with low activity. However, in agreement with previous biochemical data showing that XI is almost inactive at room temperature, the density of the active centre at 20 °C corresponds predominantly to a closed-ring glucose molecule. Conversely, at 50 °C, the electron density in the active site predominantly corresponds to an open-chain intermediate, thereby validating our prior observation and underscoring the pivotal role of temperature in the observation of reaction intermediates that are not discernible within a reasonable time frame using conventional methods [...].”

Results and discussion:

The major technical innovation of the work, the environmental control box design, is well described and very nice. The authors deserve credit for this innovation.

Figure 1:

An important conclusion is that CTX-M-14 undergo little progression in the B-factor distribution (atomic displacement parameters) from 10 °C to 40 °C and then undergo a sudden jump at 50 °C, because it has a melting temperature of 53 °C (as measured using differential scanning interferometry). By contrast, Xylose isomerase shows a steady progression in the evolution of B-factors over the entire temperature domain, but does not reach its melting temperature of 83 °C.

#8

Figure 2:

I'm afraid that I do not understand exactly what is being presented in Figure 2a and 2b. Please explain RoPE space, and what the two principle components being plotted in the graph mean physically. There appears to be some temperature dependent correlation, but it is impossible to judge the significance of this.

Figure 2c and 2d are much clearer. This representation establishes that some loop regions show larger angular distributions (ie. become more disordered) with higher temperature. This is very nice to measure directly, but seems to fall a little short of the claims of the introduction.

- We appreciate the reviewer's response and have extended the explanation of the ROPE procedure both in the main text and the supplementary material. Please refer to our response to question #1 in reply to referee #1.

#9

In the time-resolved studies, the authors build upon their own earlier work and initiate reactions using the LAMA-method. But this method is not explained in the document, rather the authors refer to earlier publications. Since the method of reaction initiation is so central to the major findings of time-resolved crystallography studies, it seems appropriate (or essential) to also describe the method of reaction initiation in this work, even if it does overlap with earlier publications. Please also state the time-scale to which this reaction initiation method is synchronized.

- We appreciate the reviewer's interest in our earlier work and have gladly extended the methods section accordingly:

“Combination of our fixed-target SSX approach with diffusion based reaction initiation was enabled via a piezo-driven droplet injector we have termed liquid application method for time-resolved analyses (LAMA). To this end protein micro-crystals are mounted into the features of the chip via a gentle vacuum to leaving small amounts residual mother liquor covering the crystals. As requirement for efficient reaction initiation highly concentrated ligand solutions (see above) are freshly prepared, sterile filtrated and degassed directly prior to the experiment. The ejection capillary (LAMA-nozzle) is placed at a distance of ca. 1 mm in front of the HARE-chip and carefully aligned to the crystal wells making use of the infrared light on-axis-viewing system. Both the humidity control stream described in the original LAMA paper as well as covering the chip surfaces with Mylar sheets to prevent crystal dehydration has become obsolete in the environmental control box and were therefore omitted during these experiments. In the LAMA approach, single droplets with a volume of 75-150 pl are shot onto each of the chip-mounted protein micro-crystals, resulting in a total ligand consumption of approximately 1.5 μ l per chip with 20,736 features. Briefly, a reaction initiation cycle proceeds as follows: (1) the chip moves into position, (2) the detector is triggered, and a t_0 image is recorded with an exposure time of 5 ms; (3) a TTL pulse triggers the droplet injector. One droplet is ejected at a velocity of approximately 2 m/s, that is a travel time of ca. 0.5 ms before it reaches the protein micro-crystal, which is taken into consideration during the post-exposure time (5 ms). (4) The crystals are exposed to X-rays and a t_1 image is recorded after the specified delay time.”

Figure 3:

This shows electron density an ensembles of structures 3 seconds after reaction initiation for extended spectrum β -lactamase and 60 seconds after reaction initiation for xylose isomerase. This is clearly the key figure against which the authors' claims that “For both systems we show that a modification of the environmental temperature modulates the underlying structural dynamics as well as the enzyme kinetics, enabling to resolve alternate structural intermediates not observed at high occupancy at ambient temperatures.”

#10

Unfortunately, the authors have tried to be too sophisticated in their presentation of panels 3a and 3d, to the extent that these become impossible to read and for temperature dependent distinctions to be made. They are trying to represent the 2Fobs-Fcalc map from sigma levels from 1.0 sigma to 3.5 sigma in the same figure and it doesn't work. Both need to be completely redrawn, as in panels 3c and 3f, which are interpretable. This failure to communicate a key figure is a major shortcoming of the work. As was shown half a century ago [R. Henderson, J. K. Moffat, The difference Fourier technique in protein crystallography: Errors and their treatment. Acta Crystallogr. B 27, 1414–1420 (1971)], difference Fourier electron density maps are much more sensitive than 2Fo-Fc maps. If there are real differences in the observations, they should be clear from difference Fourier maps. The superposition of structures does not tell me anything I can interpret in any meaningful way. The Polder omit maps (Figure 3c and 3f) are better, but it is difficult to distinguish the Polder omit map electron density reported for xylose isomerase at all three temperatures, except for the possibility that the ring-structure has slightly lower occupancy for higher temperature. The Polder omit map electron density of extended spectrum β -lactamase does show much lower occupancy at 20 °C than at higher temperature.

- According to the reviewer's suggestion to restructure the document we have split Fig3 into 2 independent figures for CTX-M-14 and XI, respectively and have added 2 additional structures at another delay time (see below). These figures now focus on maps suggested by the reviewer, while the occupancy refinement and POLDER omit maps have been moved to the supplemental material.

- We appreciate the reviewer's feedback regarding the multilevel 2Fobs-Fcalc volume representation. We hope that their educational value (the gradual shift in signal intensity) becomes apparent to the readers by combining them with conventional 2Fobs-Fcalc shown at a single constant sigma level directly below. To this end we have relied on 2Fo-Fc absolute values Electron Number Density (END) maps introduced by Holton et al. (Lang et al 2014, PNAS). As these maps are normalized to the F000 reflection, they permit a quantitative comparison between structures and thus the gain (or loss) of electron density as a function of time or temperature.
- The reviewers' critical feedback regarding the difference electron density maps was indeed most valuable. In addition to the previously reported maps, we have now included state-of-the-art difference electron density (DED) bias free Fobs-Fobs maps taking solvent masking, bulk-solvent scaling and error weighting into account (Brookner et al. 2024). As the reviewer expected, these DED maps clearly demonstrate how the electron density in the active site changes as a function of temperature. This is most notable for CTX-M-14 around the catalytic serine residue (Fig 3) and in XI around the O2 oxygen in the closed ring glucose molecule (Fig. 4), which - via the build up of negative difference electron density - demonstrate the conformational change and the catalytic turnover, respectively.
- Additional figures were included. The methods section was updated accordingly:

"[...] Fo-Fo difference electron density (DED) maps were calculated using the MatchMaps software (version 0.6.6), subtracting the apo dataset at a particular temperature, from the reaction-triggered delay-time dataset at the respective temperature according to default parameters[...]"

Thus, I am not convinced that the authors achieve the stated goal. Moreover, these electron density differences do indicate that the reaction proceeds faster at higher temperature in both cases, but this is short of the claim of "enabling to resolve alternate structural intermediates not observed at high occupancy at ambient temperatures". In both cases, the authors acknowledge: "The multiple overlapping states make an unambiguous interpretation based on the electron density alone difficult". While this is true, and the very impressive method developed is pointing towards a solution, I don't yet see clear evidence that this problem has been overcome with this method.

- To specifically address the reviewer's concern we are presenting two additional 5D-SSX structures at a delay time of 180 s at 20 °C and 50 °C, respectively (Fig. 5).
- To emphasize that under practical concerns, indeed temperature, and not time, is the critical element we have increased the delay time to 180 s, which even for poorly active enzymes should provide a sufficiently long delay to accumulate reaction intermediates or products. However, at 20 °C 180 s after reaction initiation the electron density in the active site still predominantly corresponds to a closed ring glucose structure. In contrast, at 50 °C the electron density corresponds predominantly to an open-ring intermediate. In addition to the robustness of the approach this emphasizes that catalytic key intermediates can indeed be enriched by increasing the temperature. Notably, to the best of our knowledge this is the first time that an open-ring glucose molecule was observed in the active site of XI.
- We have updated the main text accordingly:
"[...]Next, we addressed the question of whether temperature is a critical parameter for the observation of certain reaction intermediates or whether this can also be compensated for by longer delay times at lower temperatures. To this end, we compared the XI active centre at a delay time of 180 s, at 20 °C and 50 °C, respectively (Fig. 5). In practice, this delay time should be long enough to accumulate reaction intermediates even for enzymes with low activity. However, in agreement with previous biochemical data showing that XI is almost inactive at room temperature, the density of the active centre at 20 °C corresponds predominantly to a closed-ring glucose molecule. Conversely, at 50 °C, the electron density in the active site

predominantly corresponds to an open-ring intermediate, thereby validating our prior observation and underscoring the pivotal role of temperature in the observation of reaction intermediates that are not discernible within a reasonable time frame using conventional methods. [...]

Methods:

On the whole the methods are clear and report the approach to a level appropriate for scientific publication. However, the LAMA reaction initiation approach is not described in any detail, and thus the reader must access other articles to understand what was done.

- Please refer to our response to question #9

#11

Moreover, the choice of only one time-delay (3 sec in one case, 60 sec in another) has not been motivated in the article.

- We understand the reviewers' comment from an enzymological point of view. However, the rationale for selecting the current time-delays lies in emphasizing structural differences across a range of temperatures, not in delineating catalytic details.

"[...] These delay times were selected from temperature-resolved time-series of either enzyme to enable a cross-temperature comparison.[...]"

- We believe that serial crystallographic data collection at physiological temperatures, along with the ability to modify temperature during time-resolved experiments, has the potential to significantly impact structural biology. In this manuscript, the primary goal of presenting biological results therefore is to demonstrate the feasibility of the technology, illustrating that 5D-SSX, so far only pioneered in a single example, can in fact be applied to a wide range of different systems (here: mesophilic - hyperthermophilic; irreversible - reversible). The key novelty of our work lies in the, so far unique, integration of state-of-the-art TR-SSX methods (HARE, LAMA) with the ability to control temperature, *the* universal factor influencing *all* biochemical reactions. Given this focus, a comprehensive discussion of the biological implications falls beyond the scope of this technology-oriented manuscript. The structural results included here serve to validate that biologically meaningful data can be obtained, while an in-depth analysis of additional time points and biological interpretations will be addressed in future work. Nevertheless, we hope that with the revisions suggested by the reviewer, and extension of the analysis to also include two additional 180 s time-points (see above) we can convince the reviewer that the method does not fall short of its claims. We hope this clarifies our emphasis on the methodological aspects in the current manuscript and the selection of time-points and assures them of our commitment to thoroughly investigating the biological implications in subsequent studies.

#12

Line 663 refers to Sup. Fig. ??. Please correct.

- Corrected.

Summary:

This is a really nice technical innovation which is well designed and well presented. I think that the work is truly very nice and deserves publication. However, in presenting their case, the authors set the bar that the method is providing genuine new structural insight which would not be available using other classical approaches (working at lower temperature but waiting for longer time-delays for example). It is possible that a restructuring of the document to better emphasise that this goal has been achieved may be possible, especially by redrawing Figure 3, the critical figure, so as to communicate much better the key findings with clarity. This would include the presentation of difference Fourier analysis, and showing $2F_{obs}-F_{calc}$ maps with only one contour level. I would like to give the authors a chance to try to do this since the work, overall, is very impressive. But if they cannot do this, then perhaps another journal is a better option?

- We really appreciate the reviewers constructive criticism, and hope that with our revised version of the manuscript we were able to convince them that in addition to being a versatile new method that opens countless new opportunities for time-resolved crystallography also provides compelling evidence for scientific novelty. As outlined above, in our opinion scientific advance is closely linked to new research methodologies. Via this article and the detailed release of all components required for reproduction we also enable others to conduct similar experiments.

Reviewer #3 (Remarks to the Author):

#13

This manuscript describes the temperature dependence of time resolved crystallography. Unfortunately, the temperature is simply taken as the value that the sample was equilibrated to. The actual temperature is not known and it is not addressed in this manuscript. I cannot support publication of this manuscript.

- It is acknowledged that the reviewers' assertion is somewhat unexpected, given that this topic has been addressed in a comprehensive manner in the initial version of the manuscript (see quote below). As can be seen from the supplementary material (Sup. Fig. 8), the temperature inside the box has been characterised over extensive periods of time via three *independent* temperature sensors placed at three different positions in the box, one of which is directly on the chip. The discrepancy between the set-point temperature and the actual temperature has been quantified and is clearly stated in the supplementary material and also clearly visible in what is now Supp. Fig. 8, highlighted as blue shadows indicating the +/- 1°C range.

“[...] To assess how effectively different temperatures in the box can be achieved and maintained, we characterized the temperature increase from 7.5 °C to 80 °C. The data show that for both temperature control modules the humidity values quickly reach the target values. Over a temperature window of 70 °C the humidity remains stable within 2.5% of the set point. Analysis of the deviation of the chip temperature from the box temperature shows that the chip temperature follows the box temperature with a median difference of 0.7 °C, over a temperature window of 70 °C. Temperature and humidity typically equilibrate across the box and the chip within 10-15 minutes. We also examined the reliability of maintaining environmental set points during X-ray data-collection. To this end we collected X-ray diffraction data at 20 °C, 40 °C, 55 °C and 80 °C, for ca. 120 minutes and recorded temperature and humidity values in 30 s intervals during this period (Sup. Tab. 3). Remarkably, during the data collection the target humidity could be maintained within approximately 1%, while the temperature remained stable within approximately 0.5 °C. [...] In conclusion, these data show that after an equilibration time of approximately 10-15 minutes the environment in the control box has reached its target value, and can be maintained throughout extended periods of time, well beyond the typical data collection time of a chip (ca. 30 minutes). [...] “

REVIEWER COMMENTS

- Our response to the reviewers remarks are displayed in blue serif-type font.

Reviewer #1 (Remarks to the Author):

My concerns are sufficiently addressed.

Reviewer #2 (Remarks to the Author):

Report on Probing the modulation of enzyme kinetics by multi-temperature, time-resolved serial crystallography, Eike C. Schulz, Andreas Prester, David von Stetten, Gargi Gore, Caitlin E. Hatton, Kim Bartels, Jan-Philipp Leimkohl, Hendrik Schikora, Helen M. Ginn, Friedjof Tellkamp, Pedram Mehrabi.

This is a revised version of this manuscript. The authors have made sincere and respectful efforts to address the comments of reviewers in a constructive way. As I wrote in my previous report, the article motivates well the benefits of combining temperature as a fifth parameter, along with time-dependent electron density and refined atomic coordinates. In revising their article, the authors include more background to the biological systems of study, and have included data from longer time-delays to make the case that temperature is critical to optimize the occupancy of transient reaction conformations of interest. These are additions that improve the main message of the paper, as well as the accessibility of the work to the general reader. I am impressed by the technical aspects of the work overall. I am confident that the changes made to the article, and the argument made by the authors concerning the potential impact of the methods they have developed on the field as a whole, warrant publication in a broad audience journal. I therefore recommend publication of this version of the article in Nature Communications.

May I also take the opportunity to apologize to the authors for the delay in my report. The request came in during an extended period of exceptional time pressure, and I completed this report as soon as this had passed.

Reviewer #3 (Remarks to the Author):

The rebuttal does not address the concern. The temperature is not known. I cannot recommend publication of this manuscript.

- Please refer to our response to Reviewer #4

Reviewer #4 (Remarks to the Author):

The manuscript by Schulz, et al. describes the implementation of “5-dimensional” macromolecular crystallography at a synchrotron X-ray source. I note that the editor of the journal specifically requested that I provide my opinion on the accuracy of the temperature control in the reported experiments, especially with respect to X-ray beam-induced heating. Therefore, as requested, I will limit my comments to focus on this aspect of the paper exclusively.

I do think that reviewer #3 is correct that there will be some beam-induced X-ray heating of the sample. However, I strongly disagree with the notion that this is a reason to not publish the manuscript. It would be excellent to know the instantaneous temperature of the sample during the measurement; however, I think the 5D crystallography experiment described by the authors is still valuable in the absence of accurate temperature determination.

The 2024 paper by Baxter, et al. (PMID: 38848551) demonstrates that there is significant X-ray beam induced heating in serial synchrotron crystallography (SSX), which is almost certainly similar in the reported 5D experiments. In fact, in the 2024 paper, Baxter and colleagues provide a relationship between the X-ray dose and beam induced heating. It seems that this relationship could also be applied by the authors of the manuscript under review to estimate the temperature of the samples during the measurement. In the manuscript, the X-ray dose per shot is not given, so I cannot estimate myself. In this regard, I agree with reviewer #3.

The article by Baxter et al. “*Power Density Titration of Reversible Photoisomerization of a Fluorescent Protein Chromophore in the Presence of Thermally Driven Barrier Crossing Shown by Quantitative Millisecond Serial Synchrotron X-ray Crystallography*” describes a system that is based on photoisomerization of a reversible fluorescent protein. Given that neither XI nor CTX-M-14 are photosensitive proteins themselves, nor contain an optically active ligand, the approach followed by Baxter et al. to determine X-ray induced heating is not feasible in our case. Moreover, the conditions stated in the paper by Baxter et al. indicate that the authors have used an approximately 2x higher X-ray dose than we did in the current work.

On the other hand, I think it is important to bear in mind the real utility of multi-temperature crystallography, which is to alter the populations of conformational states in the crystal, so that one can observe a richer picture of the conformational landscape. The B-factor plots shown in Figures 1b and 1c, representing a shift toward more disorder at higher temperatures, illustrate that this is indeed happening in the context of the reported experiments. Therefore, the time-resolved measurements at different temperatures could reveal different substates of the molecule that are accessible during different stages of the catalytic cycle, and potentially reveal their couplings as with static, multi-temperature experiments. In my opinion, this is what makes the 5D approach promising. It is true that without knowing the exact temperature of the sample, one cannot do, e.g. an Eyring analysis, but it has also been demonstrated that reaction rates in crystallo and in solution often do not match, and therefore it is not useful to do this analysis from time-resolved crystallography experiments anyway.

Finally, I note that it is possible to measure the temperature directly from the water ring present in the X-ray diffraction images. There is substantial literature describing this in the liquid X-ray scattering field.

We anticipate that the reviewer suggested the work by Neutze and Fraser et al:

- Arnlund, D., [...], Neutze, R. (2014). *Visualizing a protein quake with time-resolved X-ray scattering at a free-electron laser*. Nat. Methods, 11, 923–926.
<https://doi.org/10.1038/nmeth.3067>
- Thompson, M.C., [...], Fraser, J.S. (2019). *Temperature-jump solution X-ray scattering reveals distinct motions in a dynamic enzyme*. Nat. Chem., 11, 1058–1066.
<https://doi.org/10.1038/s41557-019-0329-3>

In these papers the authors use the relative intensity changes of the water scattering to determine the relative temperature change within their sample. However, as the environmental control box is kept at a relative humidity of 98%, we anticipate that any changes in water scattering from the comparably small sample volume (pL scale), could be masked by the bulk water scattering within the environmental control box, making an accurate calculation on these parameters alone rather difficult.

Nevertheless, to account for X-ray induced heating within the crystals, we followed the approach recently suggested by Warren et al: (*Warren AJ, Axford D, and Owen RL (2019) Direct measurement of X-ray-induced heating of microcrystals. J Synchrotron Radiat 26:991–997*), who used an adiabatic heating model as a first approximation. Based on this model, which ignores any heat-exchange with the environment, the increase in temperature is given by the energy absorbed by the mass and the specific heat capacity of the sample. It is important to note that these estimates display an *upper boundary*, which do not take heat-exchange with the mother-liquor, to the crystalline silicon-chips, to the air-interface inside the wells, as well as convectional cooling, photoelectron escape or any energy conversion by radiolytic processes into account. That is we anticipate the actual temperature increase to be *significantly* lower.

In order to further investigate this, we have modelled the thermal diffusion process into the surrounding regions using numerical simulations (please refer to the updated Supplementary Methods). However, it is important to note that accurate simulations are highly demanding. Consequently, a conservative model has been employed to predict the spread of heat, with the thermal diffusivity of water being utilised as a lower-bound estimate, given its lower magnitude when compared to crystalline environments. The simulations, which include the diffusion of heat to the surrounding region, therefore

exhibit an over-estimation of the temperature increase. However, the calculated results indicate that the upper bound temperature increase is less than 1 K for each system.

With respect to in crystal turnover kinetics three additional aspects have to be considered:

- 1) In the present work we have used soaking times that were orders of magnitude longer (seconds to minutes) than the X-ray exposure time (5-7 ms) and hence any chance for X-ray induced heating to influence ligand binding and turnover kinetics for such slow systems.
- 2) Any potential X-ray induced heating is a constant offset to the nominal sample temperature, that is every time-point or temperature-point, respectively, is affected by the same temperature increase.
- 3) The relative temperature changes used in our work, clearly demonstrate structural differences, hence kinetic modification of the enzymes, are based on the temperature bath alone.

We note that further experimental work is required to carefully determine a more precise temperature offset. However, given the complexity of such an experiment this future work should be presented in its own regard. To take the concerns of the reviewers into account we have modified the manuscript and added another section to the supplemental material carefully lining out the statements above.

X-ray induced heating of protein μ -crystals

While the environmental control box provides a stable temperature bath for the protein crystals, their temperature is also altered by the incident X-ray beams during data collection. As laid out by Warren et al. (2019), the simplest estimate of beam-induced temperature changes is provided by an adiabatic model that ignores any heat-exchange with the environment. Thus the maximum possible increase in temperature is given by the energy absorbed by the mass and the specific heat capacity of the sample. Since the energy absorbed by the mass can also be expressed by the absorbed dose, this can be expressed as:

$$\Delta T = \frac{Q}{mc} = \frac{Q_D}{c}$$

,wherein ΔT is the temperature change, Q/m is the energy absorbed per unit mass, c is the specific heat capacity, and Q_D is the absorbed dose. The specific heat capacity of protein crystals is approximated via those described for tetragonal lysozyme (1.8×10^3 (J/ kg K)) (Kriminski et al. 2003). The absorbed dose was calculated using RADDPOSE-3D, v5 based on the parameters given in Table 2, below.

Table 1: Parameters to calculate the dose of the exposed region of the crystal in RADDPOSE-3D and calculated temperature increase.

Protein	XI	CTX-M-14
Type		Cuboid
Dimensions [μm]	20, 20, 10	15, 15, 10
Type		Gaussian
Flux [ph/s]	1.00E+12	1.10E+12
FWHM [μm]	30, 10	30, 10
Energy [keV]		12.699
Collimation Rectangular [μm]		10, 9
Wedge		0, 0
Exposure time [s]	0.007	0.005
Average Dose (exposed region) [J/kg]	36222	20234
specific heat capacity [J/kg K]	1.8×10^3 (J/ kg K)	1.8×10^3 (J/ kg K)
adiabatic ΔT [K]	20.12	11.24
non-adiabatic ΔT [K]	0.7	0.5

However, as an adiabatic model clearly ignores heat-exchange with the mother-liquor, to the crystalline silicon-chips, to the air-interface inside the wells, as well as convectional cooling, photoelectron escape or any energy conversion by radiolytic processes, we consider the adiabatic temperature increase an upper boundary, while realistic temperature changes are certainly well below these values.

Abbreviation	Quantity	Units and typical value	Reference
c_p	Heat capacity per unit mass for tetragonal lysozyme	1.8×10^3 J / kg·K	Kriminski et al 2003
α	Thermal diffusivity of water	1.4×10^{-6} mm ² / s	

In order to further investigate this, we have modelled the thermal diffusion process into the surrounding regions using numerical simulations (Suppl. Methods). To predict the spread of heat, if thermal

diffusion is taken into consideration we employed a conservative model. To this end we utilised the thermal diffusivity of water ($1.4 \times 10^{-6} \text{ mm}^2/\text{s}$) as a lower-bound estimate, compared to crystalline environments. The simulations, which include the diffusion of heat to the surrounding region, therefore exhibit an over-estimation of the possible temperature increase. Considering this more realistic scenario with heat dissipation into the surrounding region, yields a temperature increase of 0.7 K for XI and 0.5 K for CTX-M-14, respectively.

Figure 1: X-ray induced heating in protein μ -crystals. upper panel: Adiabatic model: a temperature increase of up to 20K can be observed if no heat dissipation into the surrounding medium is considered. Lower panel) Diffusive model: However, if the surrounding region is thermally conductive the temperature increase is limited to less than 1K, for each case.

With respect to turnover kinetics the delay time after reaction initiation has to be taken into consideration. While the X-ray exposure time was 5 and 7 ms, respectively, the delay times were orders of magnitude larger for both CTX-M-14 (3s) and XI (60s, 180s). This long time delay enables progression of ligand binding and turnover kinetics, irrespective of any potential X-ray induced heating.

While we note that the temperature of increase 0.5 - 1° C may further influence the turnover kinetics, it is important to emphasize that this upper boundary can be considered a constant offset that applies to all time-points. That is the deviation from the nominal temperature change would affect all time- and temperature-points uniformly, and therefore not lead to deviations from the relative temperature changes set by the environmental control box.